# The European 2015 drought from a hydrological perspective

Gregor Laaha[1], Tobias Gauster[1], Lena M. Tallaksen[2], Jean-Philippe Vidal[3], Kerstin Stahl[4], Christel Prudhomme[5,6], Benedikt Heudorfer[4], Radek Vlnas[7,8], Monica Ionita[9], Henny A. J. Van Lanen[10] Mary-Jeanne Adler[11], Laurie Caillouet[3], Claire Delus[12], Miriam Fendekova[13], Sebastien Gailliez[14], Jamie Hannaford[5], Daniel Kingston[15], Anne F. Van Loon[16], Luis Mediero[17], Marzena Osuch[18], Renata Romanowicz[18], Eric Sauquet[3], James H. Stagge[2], Wai K. Wong[19]

[1]Institute of Applied Statistics and Computing, University of Natural Resources and Life Sciences (BOKU), Vienna, Austria
[2]Department of Geosciences, University of Oslo, Oslo, Norway
[3]Irstea, UR HHLY, Hydrology-Hydraulics Research Unit, Villeurbanne, 69100, France
[4]Hydrology, Faculty of Environment and Natural Resources, University of Freiburg, Freiburg, Germany
[5]Centre for Ecology and Hydrology, Wallingford, UK
[6]Loughborough University, UK
[7]Czech University of Life Sciences Prague, Czech Republic
[8]T.G. Masaryk Water Research Institute, Prague, Czech Republic
[9]Alfred-Wegener-Institute for Polar and Marine Research, Bremerhaven Germany
[10]Hydrology and Quantitative Water Management Group, Wageningen University, Wageningen, the Netherlands
[11]National Institute of Hydrology and Water Management, Bucharest, Romania
[12]Université de Lorraine, Nancy, France
[13]Comenius University, Bratislava, Slovakia
[14]Service Public de Wallonie, Jambes, Belgium
[15]University of Otago, Dunedin, New Zealand
[16]University of Birmingham, UK
[17]Universidad Politécnica de Madrid, Spain
[18]Institute of Geophysics Polish Academy of Sciences, Warsaw, Poland
[19]Norwegian Water Resources and Energy Directorate, Oslo, Norway

*Correspondence to*: G. Laaha (gregor.laaha@boku.ac.at)

**Abstract.** In 2015 large parts of Europe were affected by drought. In this paper, we analyse the hydrological footprint (dynamic development over space and time) of the drought of 2015 in terms of both severity (magnitude) and spatial extent and compare it to the extreme drought of 2003. Analyses are based on a range of low flow and hydrological drought indices derived for about 800 streamflow records across Europe, collected in a community effort based on a common protocol. We compare the hydrological footprints of both events with the meteorological footprints, in order to learn from similarities and differences of both perspectives and to draw conclusions for drought management. The region affected by hydrological drought in 2015 differed somewhat from the drought of 2003, with its centre located more towards eastern Europe. In terms of low flow magnitude, a region surrounding the Czech Republic was the most affected, with summer low flows that exhibited return intervals of 100 years and more. In terms of deficit volumes, the geographical centre of the event was in southern Germany, where the drought lasted particularly long. A detailed spatial and temporal assessment of the 2015 event showed that the particular behaviour in these regions was partly a result of diverging wetness preconditions in the studied catchments. Extreme droughts emerged where preconditions were particularly dry. In regions with wet preconditions, low

flow events developed later and tended to be less severe. For both the 2003 and 2015 events, the onset of the hydrological drought was well correlated with the lowest flow recorded during the event (low flow magnitude), pointing towards a potential for early-warning of the severity of streamflow drought. Time series of monthly drought indices (both streamflow and climate based indices) showed that meteorological and hydrological events developed differently in space and time, both

in terms of extent and severity (magnitude). These results emphasize that drought is a hazard which leaves different footprints on the various components of the water cycle at different spatial and temporal scales. The difference in the dynamic development of meteorological and hydrological drought also implies that impacts on various water use sectors and river ecology cannot be informed by climate indices alone. Thus, an assessment of drought impacts on water resources requires hydrological data in addition to drought indices based solely on climate data. The transboundary scale of the event

also suggests that additional efforts need to be undertaken to make timely pan-European hydrological assessments more operational in the future.

## 1 Introduction

The summer of 2015 was hot and dry in many European countries when a meteorological situation similar to that of summer 2003 occurred (Van Lanen et al., 2016). The combined heatwave and drought of 2003 is known as one of the most costly

natural hazard events to have impacted Europe (EurAqua, 2004; EC, 2007; EEA, 2010; EC, 2012; García-Herrera et al., 2010). A timely analysis of the recent event of 2015 adds to the understanding of how summer droughts can develop in Europe, a prerequisite for improved drought management and policy making.

Droughts are rare events of temporary water deficit that propagate through the hydrological cycle (Tallaksen and Van Lanen, 2004; Van Loon, 2015) and affect hydrological components on various spatial and temporal scales. Drought is also a natural

hazard that affects a range of different water use sectors (Wilhite and Glantz, 1985; Gustard and Demuth, 2008; Stahl et al., 2016; Spinoni et al., 2016). Because of the complex interaction of a range of atmospheric and terrestrial processes, detailed analyses of each event are crucial to improve the understanding of the phenomenon and ultimately, the predictability of future events.

For the event of 2015, some reviews of national and regional hydrometeorological agencies already exist; these hint at its

notable severity and transboundary occurrence. For example, the Swiss BAFU published a special report reviewing the drought conditions in Switzerland, and highlighted particularly severe low flow conditions in October 2015 in the Swiss Plateau and Jura regions (BAFU, 2015). Similar reports were released for two administrative regions of western France where the drought conditions were characterised as generally moderate, even though warning levels were reached and water use restrictions came into force for some locations (l'ORE, 2015a, 2015b). Severe low flow affected navigation on major

European rivers, including the Rhine at the Dutch-German border (BfG, 2015) and parts of the Danube (Radio Romania International, 2015), but we could not find any official reports quantifying the severity of the low flow event at the time.

Because of a lack of observed pan-European near real-time hydrological data, a timely analysis of the hydrological dimension of drought events is generally challenging.

As a consequence, only meteorological (and not hydrological) indices have so far been used to describe the spatial-temporal characteristic of the drought of 2015, providing important knowledge about the droughts from a climatic perspective. In a companion paper to this study, the meteorological drought of 2015 was identified as one of the most severe droughts since the summer event of 2003, affecting a large portion of continental Europe (Ionita et al., 2017). As reported by Ionita et al. (2016), the summer of 2003 was characterized by exceptionally high temperatures in many parts of central and eastern Europe, with daily maximum temperatures 2 – 3 °C warmer than the seasonal mean (1971 – 2000). Meteorological indices such as the Standardized Precipitation Evaporation Index (SPEI) showed a dipole-like structure with rainfall deficits and extreme droughts in the central and southern part of Europe and comparatively high amounts of rainfall over parts of the Scandinavian Peninsula and the British Isles. The event of 2015, on the other hand, first appeared in the early spring in Southern France and the Iberian Peninsula, shifting toward central and eastern Europe as it developed. In August 2015, precipitation lessened the drought over SW-Europe while meteorological drought conditions persisted in eastern Europe and, notably, in central Europe where the drought lasted the longest. The most extreme climatic water deficits (precipitation less potential evaporation) were found in southern Spain, parts of France and Germany, Belarus, and western Ukraine. From a climatological point of view, the main factors controlling the occurrence and persistence of the 2015 drought event were extreme temperatures and a lack of precipitation, in turn driven by blocking episodes influenced by anomalously cold (warm) sea surface temperatures in the central North Atlantic Ocean (Mediterranean Sea).

Although hydrological drought is driven by anomalous atmospheric conditions, catchment hydrological processes may dampen or amplify the drought signal and thus, any negative drought-related impacts (e.g. restrictions to water-borne transport, degradation of aquatic ecosystems, water supply shortages, or energy production losses). Hydrologically-oriented drought studies have shown that drought in groundwater or streamflow can deviate considerably from meteorological drought in terms of lagged occurrence (Changnon, 1987; Barker et al., 2016) and statistical characteristics (Peters et al., 2003; Vidal et al., 2010; Hannaford et al., 2011; Van Loon and Van Lanen, 2012; van Dijk et al., 2013; Tallaksen et al., 2009). These differences can be ascribed to regional and local factors such as the catchments' ability to store and release water during dry weather, reflected in the amount of water stored in the soil, groundwater, lakes and snow pack, and are therefore spatially variable as well (Haslinger et al., 2014). Moreover, water managers take actions in response to the predicted impacts (e.g. on abstractions and effluent discharges, water transfers and water storage) in which hydrology plays a key role (Van Lanen et al., 2016). As such, additional analyses are warranted to better characterise the hydrological dimension of the event and mitigate its impacts.

In this study, we analyse the European drought of 2015 from a hydrological perspective based on streamflow observations. Such an analysis is challenging for several reasons. First, and foremost, the analyses require up-to-date streamflow records across Europe. The pan-European perspective is crucial to study droughts because a number of hydroclimatological processes act on the continental scale, requiring large-scale data sets to identify regional patterns. However, to date, no

publicly available pan-European databases that include near real-time records exist. Secondly, drought is a spatio-temporal phenomenon. Hence, its dynamic development over space and time, which we herein refer to as the "footprint" of a drought (e.g. Herring et al., 2015; Heim, 2015), needs to be considered. Thirdly, drought needs to be analysed by a range of indices that characterise different aspects such as magnitude or duration of the event. These indices must be comparable across European flow regimes (Water Scarcity and Droughts Expert Network, 2007). All these challenges need to be tackled in order to characterize a drought event in a manner that is relevant for a range of management purposes.

The paper stems from a collaborative effort of members of UNESCO's EURO FRIEND-Water program (IHP-VIII, UNESCO, 2012). Our study focuses on low flow events, characterised by standard methods including annual minimum discharges, drought duration, and deficit below an annual threshold (Gustard and Demuth, 2008). We analyse the dynamic development of the severity of the hydrological drought at different spatial and temporal scales and use seasonality indices to characterise the timing of key hydrological characteristics. The following research questions are addressed: (i) What is the hydrological footprint of the drought of 2015? (ii) How is it compared to the drought of 2003, often considered a worst-case benchmark? (iii) How similar, or different, are the hydrological footprints of these events contrasted to the meteorological footprints? (iv) What may be the implications of differing footprints for environmental, societal and economical drought management?

The paper is organised as follows. Section 2 describes the data collation strategy. In Section 3, we define the low flow and drought indices used in the study and present the assessment method. Section 4 presents results that characterise the event of 2015 and compare it to the drought event of 2003 at different spatial and temporal scales, based on a range of discharge and seasonality indices. We first analyse the continental scale footprint of drought events from maps of annual low flow and drought indices, and then move to a regional scale in order to elaborate specifics of drought events in more detail. The spatio-temporal development is assessed from monthly maps of indices at the pan-European scale, before analysing the "local fingerprints" of the drought from daily hydrographs at the catchment scale. Functional clustering of hydrographs was employed to put these local regimes in the pan-European context. We finally generalize our local process understanding using seasonality as an indicator of governing processes. Section 5 presents an in-depth discussion of the results, including a comparison of the hydrological footprint from all analyses with the meteorological footprint from the study by Ionita et al. (2016).

## 2 Data collation strategy

Severe droughts are characterised by a large spatial extent and may cover large parts of the European continent (EEA, 2010). Assessing the hydrological characteristics of droughts therefore requires streamflow data across Europe. However, there are still major barriers in data exchange, which have hindered initiatives to build up international data archives and to perform urgently needed transboundary intercomparison studies (Hannah et al., 2011; Viglione et al., 2010). Existing data archives such as the FRIEND-Water European Water Archive (EWA, http://undine.bafg.de/servlet/is/7413) and the Global Runoff

Dataset (http://www.bafg.de/GRDC) at the Global Runoff Data Centre (GRDC) are precious initiatives to make data accessible across Europe. But their content is still limited with respect to their spatial coverage. Moreover, they are designed as data archives of the past rather than for monitoring in near real-time. Keeping the data up-to-date is challenging, and the fact that flow records are often officially released only 2 – 3 years after recording make these archives inappropriate for a timely assessment of extreme events.

For collecting hydrological information from different European countries in near real-time, a bottom-up strategy was pursued in this study. Instead of collecting streamflow records, we collect low flow indices for approximately 800 gauges across Europe, which were calculated by partners in the individual countries. It appears easier to do the data processing in the home country and to exchange only derived data (indices), rather than the raw flow data. To ensure consistent derivation of the low flow and drought indices, we have compiled and distributed low flow software. Our software is open-source and consists of two packages based on the widely used statistical software R.

The first package, termed lfstat (Koffler et al., 2016), provides a collection of state-of-the-art functions to compute a range of low flow characteristics that are fully described in the WMO manual on low flow estimation and prediction (Gustard and Demuth, 2008). The package has been recently extended to perform extreme value statistics of both low flow discharges and drought characteristics such as duration and deficit volume. The package uses a robust approach based on L-moments to fit extreme value distributions (Hosking and Wallis, 2005). It contains approaches for pooling interrupted events (Hisdal et al., 2004) and for series containing zero values (Stedinger et al., 1993). The second package, termed drought2015 (Gauster and Laaha, 2016), builds on lfstat and extends it to perform consistent multi-station analysis. The package employs literate programing enabling all partners to generate dynamic reports that are updated automatically if data or analyses change.

We use a common reference period 01 January 1976 – 31 December 2010 to calculate indices and statistics representing long-term average conditions. The year 2015 is then compared to the characteristics of the reference period and the year 2003. As the end of available records for the year 2015 differs across countries, a common termination date (31 October 2015) was chosen.

## 3 Methods

### 3.1 Low flow characteristics

A comprehensive characterisation of hydrological drought events, such as those of 2015 and 2003, requires a number of different indices (Tallaksen and Van Lanen, 2004; Laaha et al., 2013; Smakhtin, 2001; Salinas et al., 2013). First, the magnitude of the low flow discharge is important, which may be characterised by annual minimum flows or flow quantiles with high exceedance probability. Second, the timing of low flow is important. It may be characterised by a monthly low flow index, such as the monthly 7-day minimum flow MM(7), or by a seasonality index such as the day of occurrence for the annual minimum. Third, a characterisation of drought events when the flow is below a given threshold is important. These drought events may be characterised by their duration, deficit volume, or similar indices (Yevjevich, 1967; Hisdal et al.,

2004). Each aspect may be seen as a temporal fingerprint or "signature" of the drought event (cf. Blöschl et al., 2013). From a water management perspective, these characteristics may be associated with impacts on different water-related sectors. In this study, we calculate the following range of streamflow indices to characterise the various aspects of hydrological drought.

### 3.1.1 Annual minimum discharge AM(7)

The annual minimum 7-day index, AM(7) represents the magnitude of the low flow event of a year. It is the annual minimum of a smoothed hydrograph, obtained by using a central 7-day moving average filter. The moving average filter is applied to reduce short-term disturbances of the discharge record.

### 3.1.2 Drought duration (D) and deficit volume (V)

A streamflow drought event is defined as a dry-spell in the flow record when discharge is below some given threshold (Yevjevich, 1967). Depending on the purpose of the study, different threshold concepts have been proposed. While seasonally varying thresholds (e.g. Hisdal et al., 2004; Van Loon and Laaha, 2015) enable a view on seasonal anomalies (we use them later to investigate the genesis of the low flow event and details are given in Section 4.4), our study focuses on low flow events to identify the largest absolute dry state of the system. Hence, we use a constant threshold, given by the $Q_{80}$ low flow quantile [P($Q \geq Q_{80}$) = 0.8] computed for the entire reference period. The $Q_{80}$ is used in many drought studies (e.g. Andreadis et al., 2005; Corzo Perez et al., 2011; Sheffield et al., 2009; Van Huijgevoort et al., 2014; Van Loon and Van Lanen, 2012).

During a drought event, minor precipitation events or disturbances may separate the drought event into several smaller events. As a remedy, pooling procedures have been recommended (Tallaksen and Van Lanen, 2004). In this study, the SPA (Sequent Peak Algorithm, e.g. Vogel and Stedinger, 1987; Tallaksen et al., 1997) is used. The SPA concept is based on depletion and recovery of the storage required to sustain the threshold discharge. An uninterrupted sequence of positive values of required storage defines a period with catchment storage depletion and a subsequent filling up, and two droughts are pooled if the catchment store has not totally recovered from the first drought when the second drought episode begins.

After the drought event series have been identified, the event with the largest volume per year is selected. This annual event is described by two characteristics: drought duration (D in days) and deficit volume (V in m³). As these indices refer to the most severe event per year, they represent annual maximum series.

### 3.1.3 Seasonality

The timing or "seasonality" of the low flow event may be characterised by various indices, such as onset and termination of drought (Parry et al., 2016), date of annual minimum low flow (Laaha and Blöschl, 2006a, 2006b), and others. We use here the start date ($\tau$) of the event as the most informative of the conditions leading up to the low flow event. The start date is expressed as day-of-year. To characterise the relative timing of an event, we compute the difference between the start date of

the event relative to another event, or relative to the average start date in the reference period. The relative timing ($\Delta_\tau$) is expressed in days. We further distinguish between summer (May – Nov) and winter (Dec – Apr) low flow season, and classify gauges according to their dominant low flow season into summer and winter regimes.

## 3.2 Extreme value analysis

The return period of the low flow and drought characteristics is used as a measure of their severity for a given event (here the 2003 and 2015 drought in Europe). The return periods are obtained by frequency analysis of extreme event series. For each gauging station, the estimation of return periods is performed by the following steps:

(1) Sample the annual extreme value series (AMS) from daily discharge records of the reference period. Note that low flow discharges, AM(7), represent annual minima series, whereas drought characteristics of duration and deficit volume, $D$ and $V$,

represent annual maxima series.

(2) Fit the theoretical extreme value distribution to the AMS based on L-moments. For annual minima AM(7) we use the 3-parameter Weibull distribution and for annual maxima we use the General Extreme Value Distribution as recommended in Tallaksen and van Lanen (2004). Both series might contain zero values. In case of AM(7) series, zero flows may arise due to drying up of rivers; in case of drought characteristics ($D,V$), zero values arise due to "no-drought" years, i.e. the discharge

never goes below the threshold level. In both cases a conditional probability model (e.g. Stedinger et al., 1993) is employed that takes the proportion of zero values into account.

(3) Check model fit by visual inspection of extreme value plots.

(4) Calculate the return periods of the events by inversion of their probabilities obtained from the fitted distribution.

The 2003 and 2015 events are compared using spatial plots of return periods for each low flow characteristic, and numerical

and graphical summaries. The main focus is on the return period of AM(7), a measure of low flow magnitude, but duration and deficit volumes are also investigated. The spatio-temporal development of each event is assessed based on monthly magnitudes MM(7). For comparison, the MM(7) are expressed as the corresponding return period in the annual extreme-value distribution of the entire record. Hence, the maps show in which month low flows with at least a severity of an annual low flow event occurred. Similar methods of display are used by various national and regional real-time flood and low flow

information systems that label 'hazard levels' by return periods or flow quantiles (e.g. LfU Bayern, 2016).

## 3.2 Functional clustering

Hydrographs permit the analysis of the catchments' response to the atmospheric drought signal and express "local fingerprints" of events (Section 4.4). To identify groups of catchments that show a similar hydrograph response to an event, we apply a specific form of cluster analysis known as functional clustering, which is appropriate for time graphs (James and

Sugar, 2003). Instead of considering measurements as multivariate observations, functional clustering accounts for their autocorrelation structure by considering the temporal dependency of observations. This is achieved by projecting hydrographs on a p-dimensional spline basis, equivalent to finding an adequate set of basis coefficients such that the shape of

hydrographs is well represented. In our case, a four-dimensional B-spline basis was used for the approximation. Clustering is then performed on the basis coefficients rather than on multivariate observations, which has the benefit that temporal structures are conserved. Analyses are performed using the method fscm from the R-package funcy (Yassouridis et al., 2016), which applies the functional mixed mixture model of Jiang and Serban (2012) to perform the clustering. The 2003 and 2015 events are analysed separately, based on monthly mean discharges of the Jan – Oct period. These are converted into a standardized streamflow index for each month (SSI, e.g. Staudinger et al., 2015; Barker et al., 2016) to make low flow hydrographs comparable across European regimes. For each event, the method returns a classification of hydrographs into groups of similar shape, together with an estimation of the mean hydrograph of each cluster centre.

# 4 Results

## 4.1 Continental scale footprint

Pan-European spatial patterns of low flow magnitude, AM(7), characterised by return periods $T_{AM(7)}$ are presented for 2015 and 2003 (Fig. 1, left panels), showing different extent and severity. The low flows in 2003 covered most of Europe, from central France to N-Poland and continued southeast of the Alps, with the lowest flows observed in central and eastern France, SE-Germany and E-Austria. South-eastern Europe was also affected (e.g. EEA, 2012, p.120–121), but is excluded from our quantitative assessments because of lack of data. The drought of 2015 was, within the study area, less spatially extensive and showed a contrasting response: wetter conditions in the north and south, and drier conditions in a band north of the Alps. The drought was rather moderate in most parts of this band. However, drought conditions were more severe than in 2003 in some areas around the Czech Republic, SE-Germany and N-Austria. Conditions were less extreme from E-France to S-Poland, including S-Germany and N-Romania. The lack of available hydrological data precluded any assessment of the conditions further east, but the severity of meteorological drought indices at the drought peak in August 2015 (Ionita et al., 2017) suggests that the area affected by the hydrological drought may have extended further to the east, to Ukraine, Belarus, and maybe Russia (flow in the River Don was exceptionally low pers. communication of Drs. E. Rets and M. Kireeva).

Durations of the two drought events are presented in the central panels of Fig. 1. For both events, the spatial patterns of drought durations correspond well with the low flow discharge, AM(7), patterns (left panels), but there is a clear difference in the spatial variability. Drought duration exhibits more homogeneous patterns than the magnitude of low flows in drought-affected regions. The return periods of duration are overall more moderate than for AM(7). For the 2015 event, the longest durations are observed in NE-France and SW-Germany, whereas the region around the Czech Republic is characterised by shorter drought durations. Note, however, that the results are only preliminary as drought may not have concluded everywhere by the end of the records (October 2015), as we further discuss in Section 5.1. For the 2003 event, the longest drought durations are observed for S-Germany and NE-Austria, whereas central and eastern France exhibit shorter durations. Again, the 2015 event appears to cover a smaller portion of the study area than the 2003 event, but in the affected regions, the return periods of duration are comparable to those of 2003.

The deficit volume is a cumulative measure of drought that integrates information on both flow magnitude and duration. In 2015 deficit volumes with return periods of 50 years and more occurred (Fig. 1, right panel), with the largest deficits occurring in S-Germany, west of the area with lowest flows (Fig. 1, left panels). In regions where the drought event was short, such as central France and NE-Austria, deficit volumes are small regardless of AM(7). For drought-affected areas, deficit volume exhibits a rather patchy pattern similar to pattern of low flow discharge AM(7), and thus reflects local hydrological conditions. Compared to 2003, the 2015 event covered a smaller part of the study area in terms of deficit volume, but with high severity within the drought-affected region.

## 4.2 Comparison by regions

The severity of the events of 2015 and 2003 is compared for three contrasting regions: the Czech Republic, which was the most affected region in 2015; E-France (Rhine and Saône hydrographic regions), which was one of the most affected regions in 2003; and S-Germany (Baden-Württemberg and Bavaria south of river Maine), which was strongly affected in both years. Figure 2 shows the distribution of annual minima low flow, AM(7), and drought deficit volume for each event and region. Catchments with a winter low flow regime are presented in separate box-plots, as low flows that occur in winter may be triggered by different processes (Van Loon et al., 2015). Overall, the regional distribution of return periods are broader for AM(7) than for deficit volume, and AM(7) shows higher return periods in the most affected catchments.

There is a major difference in the severity of the events of 2003 and 2015 in the Czech Republic (left panels of Fig. 2) with moderate return periods in 2003 for both low flow discharge AM(7) (median $T_{med}$ = 10 yr) and drought deficit volume ($T_{med}$ = 21 yr). In 2015, deficit volumes have a slightly higher severity ($T_{med}$ = 32 yr), whereas record low values of AM(7) were observed, leading to return intervals of more than 100 years in more than half of all catchments (i.e., $T_{med}$ > 100 yr). Note that due to the limited record length (around 35 yr), estimated return periods are only indicative and must be interpreted as such.

In E-France, the 2003 event was characterised by high return periods for AM(7) (often $\geq$ 100 yr) and moderately high severity in deficit volume ($T_{med}$ = 27 yr). In the pan-European context, it was one of the most severely affected regions in 2003. In 2015, however, the drought was relatively mild in terms of AM(7) (only marginally below the average summer conditions), but slightly more severe in terms of volume.

S-Germany, located between the Czech Republic and E-France, experienced similar severity for both events (summer catchments), with slightly higher return periods in 2003 than in 2015. Both events were more severe in terms of deficit volumes ($T_{med}$ = 26 yr in 2003, and $T_{med}$ = 19 yr in 2015) than in terms of AM(7) ($T_{med}$ = 13 yr in 2003, and $T_{med}$ = 9 yr in 2015). The S-Germany region appears to be prone to relatively long drought periods that let deficits accumulate over a long time. Note that analysis of catchments with winter low flow regimes does not show exceptionally dry conditions in the winter prior to the 2015 summer low flow event.

Finally, when comparing across regions, it can be seen that the return periods for low flow magnitude associated with both events vary gradually between the values in E-France and those in the Czech Republic, suggesting a gradient of low flow

magnitude increasing from east to west in 2003, but increasing from west to east in 2015. The deficit volumes show a similar, but less pronounced gradient in 2015, and there is almost no gradient for the volumes in 2003. Again, the less pronounced gradients for deficit volumes are due to the long durations of drought events that make deficit volumes less dependent on peak magnitude.Summary statistics of streamflow drought characteristics for individual countries in Europe are provided in Appendix A.

### 4.3 Spatio-temporal development

Figure 3 shows the spatio-temporal development of the 2015 and 2003 drought events based on each flow record's low flow discharge, MM(7) (monthly magnitude) from February to November. Table 1 provides a statistical summary of the affected stations. In our results, the maps for both drought events show that an exceptional situation started to develop in June (with first indications of an anomaly appearing already in May) when discharges began to fall under the average annual low flow discharge. However, onset was more dramatic in 2003, affecting a larger region more quickly and homogeneously. Interestingly, the regions that were first affected during either event are consistent with the regions that were later also affected most severely, i.e., central France and E-Austria in 2003, and Czech Republic and central Germany in 2015. By the end of July (2015) or beginning of August (2003), the full spatial extent of both droughts was reached. This is also reflected in the monthly number of stations experiencing drought (ref. Table 1). During this "peak" of the drought, differences emerged with respect to the drought characteristics and recovery periods. In 2003, the peak of the drought was reached in August, with a clear recovery visible in September, when the most affected regions returned to more moderate conditions. The spatio-temporal development seen from MM(7) is consistent with the findings of Stahl and Tallaksen (2010) where the drought event was assessed from daily discharge snapshots based on EWA stations. The recovery started in western Europe, reached the region north of the Alps in October, and finally affected eastern Europe in November. At this time, most parts of Europe had returned to above-average low flow conditions, except for a band north of the Alps (UK to Poland) that remained under mild drought conditions. In 2015, the timing of the peak differed across regions from August to October (some regions around S-Germany remained under moderate drought conditions by the end of October 2015), with recovery starting later. This recovery was first seen in the western regions and was slower than in 2003.

### 4.4 Local fingerprint

The analyses so far have shown that a band north of Alps was most affected by the 2015 drought, but the severity of the event differed between regions. Overall, this band corresponds well with the region affected by meteorological drought during the peak of the event. However, there are important differences between hydrological and meteorological drought patterns on a smaller, regional scale. For example, at the eastern end of the Alps, N- and E-Austria exhibit similar precipitation anomalies, temperature anomalies, and SPEI3 values for the summer drought season (JJA) (Ionita et al., 2017). Nevertheless, there are striking differences in low flow discharges and volumes. In the following, we look closer into why some catchments produced very low flows in 2003, but not in 2015.

To gain insight into these differences, we selected hydrographs from both events in two contrasting catchments (Fig. 4 and 5). The first example is the gauge Altschlaining at river Tauchenbach in E-Austria (Fig. 4). The catchment has an area of 89.2 km² and the altitude of the gauge is 316 m.a.s.l. Its geology consists of phyllite and schist in one part, and clay marl and sandstone formations in the other part of the catchment. The gauge represents a region that fell extremely dry in 2003, but exhibited no severe low flows in 2015. Figure 4 shows that the reason for the contrasting behaviour can be found in the conditions preceding the summer event. In 2003, discharge was already lower than normal during the winter and spring season. This is clearly indicated by the hydrograph, which started to decrease below the seasonal $Q_{80s}$ threshold in winter. The seasonal deficits steadily increased during spring, leading to an early onset of the low flow event in the beginning of May. The meteorological summer drought exacerbated the hydrological situation and yielded the lowest discharges since the beginning of streamflow records. In 2015, the meteorological situation in summer was comparable to 2003 in this region, with small inter-annual differences of accumulated summer (JJA) precipitation (Fig. 11e of Ionita et al., 2016). However, the hydrographs started at a much higher level in 2015, pointing to very wet preconditions. This is also reflected by the higher January-SPI6 value of 1.5 in 2015 as compared to 1.1 in 2003 in this region, with January-SPI6 accumulating August to January precipitation, and values above / below zero indicating wet / dry anomalies (Ionita et al., 2017). Streamflow remained above the average seasonal regime until June (indicated by the $Q_{50s}$ line), leading to a late onset of the low flow event in August. It appears that surplus water from the winter and spring seasons fed discharge during the summer drought and thereby prevented an even more extreme low flow event from developing in 2015.

A different situation occurred in N-Austria at the Imbach gauge on the river Krems. This catchment has an area of 305.9 km², the altitude of the gauge is 231 m.a.s.l., and its geology consists of granite and gneiss. This river fell only moderately dry in terms of flow magnitude in both 2015 and 2003. However, Fig. 5 shows contrasting recession behaviour in the hydrographs, again due to different preconditions. The 2003 low flow event is characterised by an almost uniform streamflow recession. The Imbach catchment is situated in a region that was heavily affected by the flood event in August 2002, reflected by an extremely high January-SPI6 value of about 4.0. Streamflow started to decrease in winter, but remained above seasonal average conditions ($Q_{50s}$). By spring, discharge was close to seasonal drought conditions, but there was enough stored water in the catchment to sustain streamflow through summer 2003. The 2015 low flow event, on the other hand, was characterised by much drier preconditions in winter. However, there were several rainfall events in spring, which can be seen in the number of pronounced streamflow peaks. These precipitation events delayed streamflow recessions and prevented more severe low flows from developing during the summer of 2015.

## 4.5 Spatial clustering of hydrographs

To generalise the findings obtained from interpreting seasonal anomalies of the two hydrographs to the European scale, we performed functional clustering of hydrographs across Europe. Results for 2003 are shown in Fig. 6. Altschlaining, east of the Alps, is one of the driest regions in a band between central France and eastern Austria (belonging to Cluster 3). The cluster is characterised by an early onset and severe dry anomalies. Imbach, north of the Alps, belongs to a cluster of stations

with much smaller anomalies (Cluster 4). The distribution of both clusters corresponds well to the patterns of the most affected and moderately affected regions, respectively, across Europe. The region around southern Germany forms a distinct cluster (Cluster 5) with wet anomalies in spring leading to later recessions than in the surrounding area. The late onset of the drought in 2003 can be well explained by high precipitation amounts in the summer preceding the event, causing major floods in this area in 2002. The anomalously wet area is clearly visible in the January-SPI6 map in Fig. 8 (left panel) that shows standardized precipitation anomalies of the August – January period. These wet preconditions explain why the region is behaving differently than the surrounding regions despite that experiencing similar meteorological drought conditions in summer.

Clustering for the 2015 drought event is shown in Fig. 7. Imbach belongs to a cluster that contains the most affected catchments, situated in the band ranging from central France to Czech Republic (Cluster 1). Altschlaining belongs to a cluster that exhibits a similar, but somewhat delayed signal, showing a tendency towards less severe conditions (Cluster 6). Catchments in this cluster are characterised by a later onset, lower magnitude and earlier termination of the drought. Southern Germany deviates from these general patterns and forms a distinct group (Cluster 2). The cluster shows striking similarities with Cluster 5 of the event of 2003. The region is characterised by a later onset, moderate low flow magnitude and a later termination of the streamflow drought.

## 4.6 Effect of preconditions

The previous section has shown that meteorological drought indices such as the January-SPI6 (ref. Fig. 8) contain relevant information about the preconditions of a hydrological drought event. Here we investigate the extent to which spatial patterns of standardized precipitation and streamflow seasonality indices that quantify preconditions can explain drought magnitude at the pan-European scale. While some regional features of the hydrological drought of 2003 can be well explained by a superposition of January-SPI6 (Fig. 8, left panel) with August-SPEI3 from (2017), this does not appear to be valid for the event of 2015 (Fig. 8, right panel). The region around the Czech Republic was most affected in 2015. However, most parts of this area exhibited only moderate meteorological drought conditions in summer (August-SPEI3 = about -0.5 – -2.0) and above-average (i.e. wet) preconditions (January-SPI6 = about 0.2 – 1.2). Accordingly, the antecedent half-year precipitation (represented by the SPI6) may not be considered a skilful indicator of antecedent catchment wetness in all cases.

Preconditions are also reflected by the timing (seasonality) of the onset of the drought event (Section 4.4). Figure 9 shows the relative seasonality (day of occurrence $\Delta_\tau$) of the onset of events across Europe. Overall, the spatial patterns of the 2003 event (Fig. 9, left panel) and the 2015 event (Fig. 9, right panel) show much earlier onsets than the long-term average onsets. The early onset increases the risk of a severe low flow event to develop during an extreme meteorological drought. In 2003, the widespread early onset explains the larger scale of the drought event in 2003. Secondly, the spatial distribution of catchments experiencing early / late onset corresponds well with the most / least affected regions across Europe. In 2015, the hydrological drought was most severe in a band north of the Alps, an area that had a notably early onset. In 2003, the same region encountered a later onset and accordingly, more moderate low flow conditions. The most affected area in 2003,

central France and E-Austria, similarly experienced a very early onset. Thirdly, it is interesting to analyse spatial patterns of the relative onsets for both events (Fig. 9, central panel). The patterns appear closely related to the relative severity of events in terms of magnitude (Fig. 1). The affected band across central Germany is marked by reddish colours, indicating a somewhat earlier onset and more severe low flows in 2015 than in 2003. Central France and the band along the pre-Alps crossing S-Germany is marked by bluish colours and a less severe drought event in 2015. The same indication were given for the part of southern Europe covered by the study, which shows blue colouring in 2015 and, thus, suggesting less severe low flow conditions.

We finally quantify the predictive skills of standardized precipitation indices of different accumulation periods (6, 9 and 12 month) and hydrological onset index for low flow magnitude, by correlating their spatial patterns with those of low flow discharge AM(7). As we are focusing on summer events, we conduct the analysis based on stations with a summer low flow regime. For the two events 2003 and 2015, the relative onset $\Delta_\tau$ exhibits a negative Spearman correlation coefficient with AM(7) of -0.55 and -0.52 (earlier onset implies more severe low flows). These correlations are much stronger than for the January-SPI6 (0.25 and -0.03), where the positive correlation found for 2003 appears spurious (since it would indicate that a high SPI6, representing wet preconditions, would lead to more severe low flows). Weak correlations are also observed for SPIs with longer aggregation scales. For instance, April-SPI9 correlations are 0.19 and -0.10 (with similar correlation coefficients of SPI9s and SPI12s for January to May). Only SPI6-values of April and May, reflecting winter and spring precipitation, are somewhat better correlated with the low flow magnitude (values of -0.03 and -0.34 for 2003, and -0.28 and -0.17 for 2015).However, the correlations are still much lower than those of the relative onset $\Delta_\tau$.

## 5 Discussion

### 5.1 Merits and limitations of the study

This study presents a first timely analysis of the 2015 hydrological drought at a nearly pan-European scale. Drought is one of the most costly hazards as it affects a number of water-related sectors. The potential for damage is high as drought events typically affect large areas and may last for a considerable period of time. Although mitigation measures on a European or regional level require timely and accurate information about the physical system, a timely analysis of the hydrological situation is challenging at the continental scale. As described in Section 2, there are major barriers of data access, especially for eastern European countries. Wherever data are available, compatibility poses a challenge. All these obstacles were overcome in this study by capitalizing on the potential of a well-established international network, provided by UNESCO's EURO FRIEND-Water program. Without this network, a timely analysis of the event would not have been possible. The resulting collated dataset offers a unique opportunity to analyse the 2015 drought from a hydrological perspective across most of Europe.

Despite these merits, the study has several limitations. For various reasons, we could not cover the European continent as a whole. Hence, there are blank spots in the south, east, and south-east Europe that could not be filled because data were not

available. Results were also not available where streamflow is not a meaningful drought indicator, such as for intermittent rivers in semi-arid regions and for highly regulated river systems. For these areas, meteorological data are currently the more useful and readily available information. Climate indices, such as SPEI3 for August, suggest that the south and south-east experienced relatively wet conditions in 2015, so the lack of information in those regions likely had only minor consequences for the results of the hydrological drought analysis. For 2003, however, it was the south-eastern parts of Europe that were particularly affected, and to gain full insight into the footprint of the hydrological drought would require a larger effort in obtaining data exchange for his region.

A second limitation arising from the need for a timely assessment is that limiting analysis until the end of October 2015 might not have captured the true end of the drought for some sites. For example, in the major rivers Rhine and Danube,some gauges in SW-Germany and the northern pre-Alps of Bavaria and Upper Austria suggested that discharge was still decreasing after 31 October (additional analysis not shown). Large catchments are known to respond slowly to atmospheric signals due to large storages and delay processes in the river network (Gustard and Demuth, 2008; Laaha et al., 2013; Salinas et al., 2013), explaining the later termination date for these gauges during the 2015 drought. Catchments in the northern pre-Alps are much smaller and typically fast responding, reacting much quickly to autumn precipitation. Further, summer low flows are often followed by frost in these regions, which imply that the low flow situation continues into the snow season. The lowest flows typically occur in October and November in these regions (Laaha and Blöschl, 2006b) and thus, it may happen that the low flow situation was even more severe than reflected by the drought characteristics calculated in this study. Figure 10 shows the stations that are still under drought, according to the SPA method (i.e. storage has not totally recovered from the summer drought) at the end of the study period (31 October 2015). For these stations, the analysis of low flow magnitude and deficit characteristics (duration and volume) for 2015 may be incomplete. Duration is notably sensitive to the further development of the drought situation, as it will grow linearly over time until the termination of the event. Volumes are more robust since their accumulation over time also depends on the magnitude of streamflow, and it was shown above that streamflow was already increasing at the end of end of records for most gauges. As a consequence, we consider the results for low flow magnitude in this study to be quite representative for the full 2015 event, and the results on deficit volume to be more representative than drought duration, which remain useful for relative comparisons between gauges.

## 5.2 Hydrological vs. climatic footprint

Drought events are often described either from a climatic or from a hydrological perspective. We are interested in how well these perspectives agree in terms of key drought characteristics. For both perspectives, it is common to describe the drought (location, extent and severity) at the peak of the event, when conditions are most extreme. This phase of the drought is important because it is the time when we expect most impacts. From the atmospheric perspective, the peak of the event can be gleaned from a meteorological index such as a (daily or monthly) running SPEI3, using an accumulation period that matches the usual lagged response time of catchments (Haslinger et al., 2014; Stagge et al., 2015; Ionita et al., 2016). From a hydrological perspective, the flow is at its minimum when the catchment water balance is also at its minimum. Comparing

AM(7) (ref. Fig. 1) and SPEI3 (ref. Fig. 3 of (2017)) for the 2015 event, we see that differences are small. Thus, the climate and hydrological footprints are similar (in size and location), but there are some regional deviations in the two patterns. For example, in the Czech Republic,  low flows were among the most severe for Europe, but SPEI3 was only moderately dry.

Analysing the dynamic development of the drought reveals much greater differences. Ionita et al. (2016) showed that from a

climatic perspective, the 2015 event first appeared in S-France and the Iberian Peninsula, where dry anomalies developed during spring (May and earlier). The meteorological drought then shifted slowly to western Europe and along the northern Alps to the east. Although no streamflow data from the southern Iberian Peninsula were available for our analysis, it appears that the hydrological event had a somewhat different dynamic, with first appearance of low flows in the Czech Republic and central Germany, followed by an extension to the south, west and east. This difference in the spatial development of

hydrological and climatic drought is mainly due to the role of the catchment in transforming the climatic signal over different time scales. The analysis of streamflow dynamics at the catchment scale showed that preconditions for spring, winter, and earlier  are critical in controlling the temporal and spatial development of summer streamflow droughts, in combination with storage and release properties of the catchment. This is in line with Van Loon and Laaha (2015) who emphasized that the spatial variation of hydrological drought severity is highly dependent on terrestrial hydrological

processes, among which preconditions and storage play an important role. The magnitude of discharge in winter and spring reflects the initial condition of the catchment before the start of the dry period. It is the combined effect of the initial condition and catchment processes superimposed with the atmospheric signal that explains the development of a hydrological drought.

### 5.3 Implications for water management

Our study presents drought as a complex phenomenon that leaves different footprints on land and atmosphere. Because of the complex interaction of atmospheric and land processes, each event is unique and adds to the current knowledge on drought generating processes. We performed a comparative assessment of the 2015 drought event relative to the benchmark event of 2003 to gain new insight from their similarities and differences. Our findings may contribute to drought-related water management in several ways.

First, the importance of winter and spring (pre-)conditions is crucial for early warning and risk assessment. Wet preconditions caused by precipitation in spring, winter and even in earlier periods (such as the extreme rainfall event of August 2002 for the 2003 event in some regions) can substantially modify the climatological drought signal and thus, the development of a hydrological drought. This was clearly demonstrated by the example of the Imbach catchment (for both events) and the Altschlaining catchment (for 2015), where water from stored sources sustained streamflow and prevented

more severe streamflow droughts. Wet (or dry) preconditions can also lead to land-atmosphere feedbacks, e.g. increased (decreased) soil moisture leading to a higher (lower) probability of precipitation that can relieve or amplify a severe drought (c.f. Seneviratne et al., 2010). For catchments with substantial storage, an extreme streamflow drought could only develop following dry preconditions, such as the case for the Altschlaining catchment in 2003. For early warning and prediction, an

early detection of drought-fostering conditions is therefore crucial. Our study suggests that a regional mapping of spring discharges in "hazard maps" (in the spirit of Fig. 3), and the relative seasonality of the beginning of an event with respect to average and benchmark conditions (such as in Fig. 9), can offer valuable information for early warning and detection of potential drought-affected regions. Based on the events analysed in this study, the regions that were later most affected by the drought could be identified by unusually low spring discharges and an early onset of the low flow event. One may expect that similar results would be valid for other extreme events as well, but this requires additional studies. Further detailed studies of the link between climatological drought and hydrological drought characteristics, including their dynamic behaviour, may contribute to improved models and better-informed decisions in water management.

Secondly, most drought impacts are not simply caused by a lack of rainfall, but more so by a lack of available water resources at the proper time, whether it be seasonal anomalies of soil moisture, groundwater, streamflow, or sometimes by a direct effect of heat exposure.

During the drought of 2015, many impacts were noted. Besides the widespread agricultural losses due to the meteorological and soil moisture drought, Van Lanen et al. (2016) also described a range of impacts that were directly related to streamflow drought. They include deterioration of water quality and instream habitats for fish, violation of legal minimum flow requirements, impairment of river navigation, reduced energy production from hydropower and thermal power plants, and water supply restrictions related to lack of reservoir inflows or to aquifer infiltration. For example, the French waterway network authority (vnf) reported on restrictions for navigation in some canals in north-eastern France from mid-June onwards (vnf, 2015a) and had to close some canals in mid-July (vnf, 2015b). As early as the beginning of June, deterioration of surface water quality was reported in the Netherlands and one month later in Germany. The Drought Management Centre of South Eastern Europe (DMCSEE) reported primarily agricultural drought impacts in its monthly monitoring summaries for the region during early summer, but from July onward hydrological drought impacts were mentioned (DMCSEE, 2015). To study the link between drought magnitude and impacts, some studies have used retrospective collections of reports that were coded into occurrence of impacts in particular sectors and categories. An example is the European Drought Impact Inventory (EDII, http://geo.uio.no/edc/droughtdb/) (Stahl et al., 2016), which has been used to describe the impacts of previous droughts, including the event of 2003 in detail (e.g. Stagge et al., 2013). A similar impact report collection for the 2015 event is currently in progress, but not yet available as the EDII is only a research project and no operational effort exists, and each report must be carefully handled and the coding cross-checked manually.

Thirdly, in drought management, different kinds of indices at various temporal scales have been considered (Bachmair et al., 2016). Crops in different growth periods differ in sensitivity to heat stress and lack of rainfall. Hence, accurate predictions of the timing and magnitude of meteorological drought and heat waves are key when one aims to optimize irrigation water. Hydrological indices, on the other hand, are relevant for a number of other water management tasks, such as those related to hydropower and navigation, but also for water quality, aquaculture and in-stream ecology. For the latter, low flow discharge during summer heat periods is critical, as high solute concentrations at higher temperature may yield a cascade of hydrochemical processes with adverse effects on water quality. For navigation, the duration of critical water levels is

important, whereas for hydropower the total deficit over the event determines the economic losses. In the absence of groundwater data, deficit volumes (in addition to baseflow and recession analysis), representing the reduced outflow of stored sources in the catchment, may also be indicative for groundwater resources in a way that is relevant for water supply and irrigation planning. All these types of drought impacts mentioned above occurred in 2015. The German Federal State of

Bavaria reported violations of the oxygen concentration threshold in rivers (LfU Bayern, 2015). Switzerland and southern Germany issued restrictions to a common law that normally allows citizens to extract small amounts of water from rivers to water their gardens (BAFU, 2015 and exemplary: Stadt Waldkirch, 2015). Impacts on the navigability of larger rivers and thus on waterborne transport were first reported for the Elbe, Weser, and Odra Rivers in late May, for the River Danube from mid-late July onwards, and for the River Rhine from early August onwards (BfG, 2015). From September onwards, springs

dried up in the mountain regions of S-Germany affecting local water supplies (personal communications).

Characterising events in a way that is relevant for drought management requires timely pan-European data to be made publically available. Such data platforms exist for meteorological variables, but similar structures for the exchange of hydrological data are missing and need to be established. A lesson learned from this study is that droughts need to be characterised, monitored, and understood from both a hydrological and climatic perspective, implying that it is not sufficient

to analyse only meteorological or climatological drought indices to learn the full range of impacts on the natural, social and economic system. Current research is fragmented across different disciplines with partly different perceptions, with studies focusing on either the atmospheric or on hydrological perspective. In our collaborative effort within the EURO FRIEND-Water network, we aimed for an integrated view from climatologists and hydrologists across several countries (pan-European scale). This approach fostered the exchange of ideas, and enabled additional insights into the interaction of

atmospheric drivers and catchment processes across regions that would not have been studied unitedly otherwise. This is especially important when investigating a large-scale phenomenon such as drought, which produces adverse effects on several components of the hydrological cycle. There are indeed a vast number of open questions related to drought that require interdisciplinary research. For instance, a better understanding of how drought propagates through the water cycle would profit from exchanging specific knowledge and data about drought processes and about how to best characterise them

by indices. A dialogue between disciplines may yield improved indices that are better suited for understanding drought dynamics across scales. In addition, there is a need for indices that address operational needs and that are relevant for predicting drought impacts. Another gap in our knowledge that requires interdisciplinary research is the role of land-atmosphere feedbacks in drought generation. Such feedbacks may be important to understand the persistency of events and contribute to the development of both climate models and hydrological models.

All these examples demonstrate that a more complete understanding of droughts would be beneficial for a range of water management tasks, which also applies to drought policy making. Yet, a holistic view of drought is hampered by fragmentation into several disciplines. Communities need to collaborate closer to further enhance our understanding of hydrometerological drought.

## 6 Conclusions

In this study we analysed the European drought of 2015 from a hydrological perspective. In a unique community effort of data collection and processing according to a common protocol, the analysis was based on a range of low flow indices calculated from observed streamflow records of approximately 800 gauges across Europe. Thus, it provided the first insight into the spatial and temporal characteristics of the hydrological drought of 2015. With a contrasting response of wet conditions in the north and south, and dryer conditions in a band north of the Alps, spanning from E-France to S-Poland and N-Romania, the hydrological drought had a different spatial extent than the benchmark drought of 2003. In terms of low flow magnitude, the drought was rather moderate in most parts, but severe in a focal area from Czech Republic, SE-Germany and N-Austria, with return periods of more than 100 years. Here, the event was even more severe than the event of 2003. In terms of deficit volumes, the drought was particularly severe in a region around S-Germany where the duration of the event was notably long.

The data also revealed an interesting dynamic development of the hydrological drought with a southward spread and expansion from spring to summer and autumn. This development differs from the clear west-to-east spread of the climatological drought (Ionita et al., 2017). The difference in spatio-temporal characteristics of the climatic and hydrological drought can best be explained by diverging preconditions in the catchments. Hydrographs provided local fingerprints of drought processes in which we found evidence that extreme droughts emerged as a consequence of dry preconditions in the preceding winter and spring months. Where wet preconditions occurred, low flow events and thus the onset of drought developed later, and the event was overall less severe. The preconditions can be well described by the onset of the hydrological event, which is notably higher correlated with the severity of the two events than with a long-term meteorological drought index such as SPI6 and SPI12. Overall, preconditions seem to control the geographical patterns of onset, scale, and severity of the drought within the different regions studied. Moreover, the focal region of the drought event coincides with the region with the earliest onset.

The results of this study demonstrate that drought leaves different footprints on the various components of the water cycle, on different spatial and temporal scales; with hydrological drought as a superposition of preconditions and the atmospheric water deficit in summer leading to the extreme streamflow drought in 2015. Hence, there is need for developing effective indicators and indices to detect and assess drought situations throughout Europe, as indicated by Water Scarcity and Droughts Expert Network (2007). Hydrological drought events differ from meteorological events because catchments collect and retain precipitation water, which exerts a modulating and delaying effect on meteorological water deficits. This finding has implications for the prediction and management of the impacts of hydrological drought, which the event of 2015 illustrates in a multitude of ways. Using a single meteorological drought index such as SPEI may not suffice as a drought indicator in this respect. For many sectors suffering from long-term accumulated deficits, streamflow and groundwater hydrological indices are likely more relevant. A more targeted large-scale drought monitoring, however, requires hydrological data on a pan-European scale. Such data is available to some extent (ref. European Drought Observatory

([http://edo.jrc.ec.europa.eu](http://edo.jrc.ec.europa.eu))), but largely unavailable for free and in near real-time. Providing the necessary data for managing drought in a pro-active way requires a concerted action of the hydrological and climatic communities. Such action should include pan-European provision of monitored streamflow and groundwater data in real-time or near real-time, of hydro-meteorological variables, and of multi-monthly and seasonal forecasts for both climatic and hydrological variables (Van Lanen et al., 2016). The results also highlight the need to implement national and European water policy where additional efforts are undertaken to make near real-time hydrological data available across borders in order to make drought management more operational in the future.

## Acknowledgements

This study was conceived by a team of European drought experts from the UNESCO EURO FRIEND-Water Low Flow and Drought network, which enabled collection of near real-time hydrological data and impact reports across Europe that otherwise, would have been impossible. This research supports the work of the UNESCO-IHP VIII FRIEND-Water programme. Data provision by national hydro-meteorological services was highly appreciated. Funding from ACRP project DALF-Pro (GZ B464822) is gratefully acknowledged. The paper is dedicated to Dr. Alan Gustard (Institute of Hydrology / Centre for Hydrology and Ecology, Wallingford, UK), one pioneer of transboundary low flow research who inspired this community effort to overcome data barriers across Europe and beyond.

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

Table 1. Statistical summary of stations under drought for the individual month of 2015 (top) and 2003 (bottom). $n_d$ is the number of stations under drought (i.e. with a severity of an annual low flow event and more). Median and quartiles summarize the return periods of low flow discharge MM(7) (monthly magnitude) of these stations (expressed as the corresponding return period in the annual extreme-value distribution of the entire record).

|                | Jun   | Jul   | Aug   | Sep   | Oct   |
|----------------|-------|-------|-------|-------|-------|
| 2015           |       |       |       |       |       |
| $n_d$          | 78    | 261   | 332   | 293   | 227   |
| Lower quartile | 2.42  | 2.58  | 2.67  | 2.83  | 2.66  |
| Median         | 2.86  | 3.38  | 4.13  | 4.51  | 3.87  |
| Upper quartile | 8.56  | 6.49  | 11.86 | 12.45 | 8.75  |
| 2003           |       |       |       |       |       |
| $n_d$          | 169   | 353   | 527   | 486   | 318   |
| Lower quartile | 2.32  | 2.45  | 3.37  | 2.98  | 2.62  |
| Median         | 3.00  | 3.43  | 6.46  | 4.93  | 3.52  |
| Upper quartile | 4.26  | 6.31  | 17.00 | 9.99  | 6.26  |

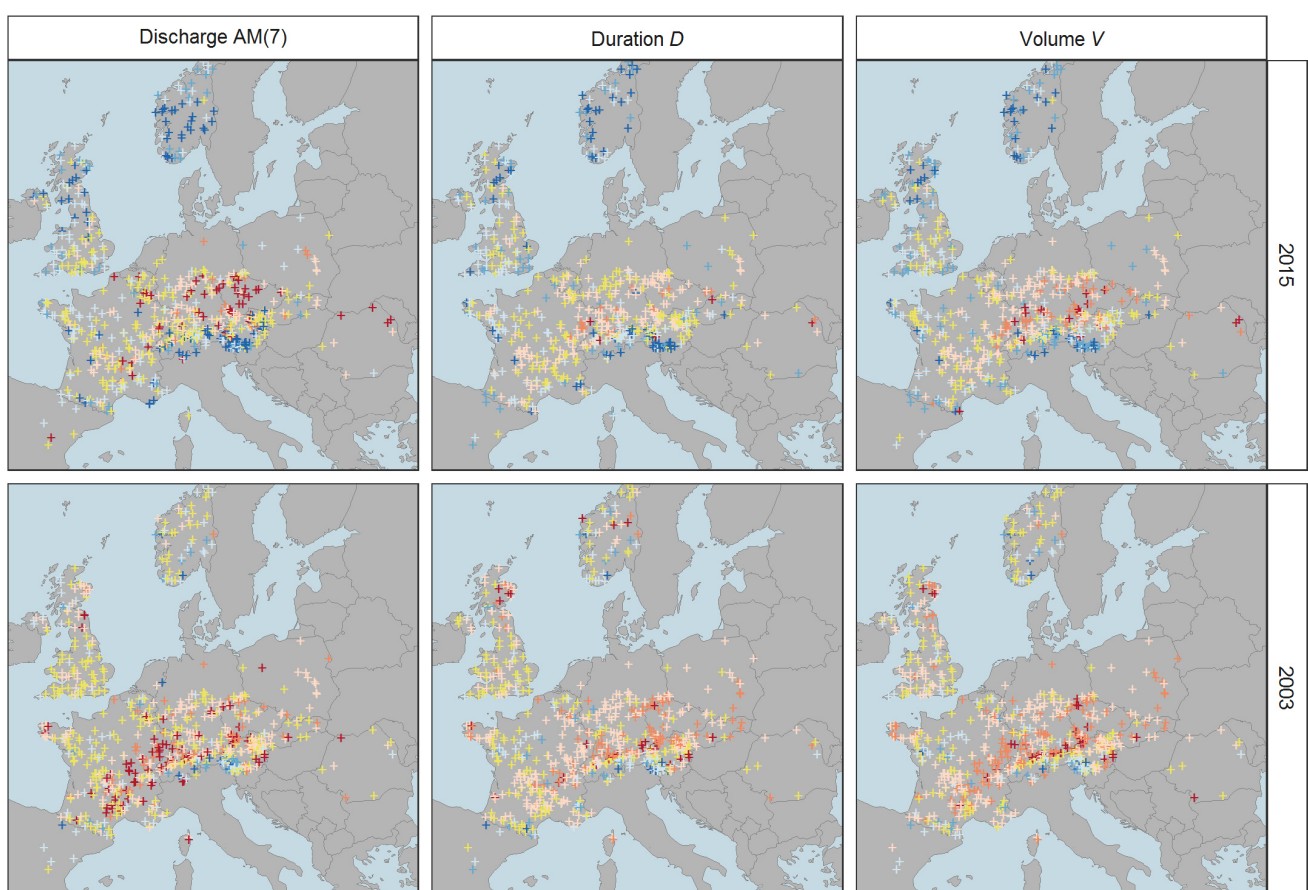

Figure 1. Return periods $T$ (in yr) of annual low flow discharge AM(7) (left), duration $D$ (centre) and deficit volume $V$ (right panels) for the drought events of 2015 and 2003. Low flows and drought conditions below average conditions (return period > 2 years) are indicated by yellow to reddish colours. Severe events (return periods (20,50] and > 50 years) are indicated by orange and red colours.

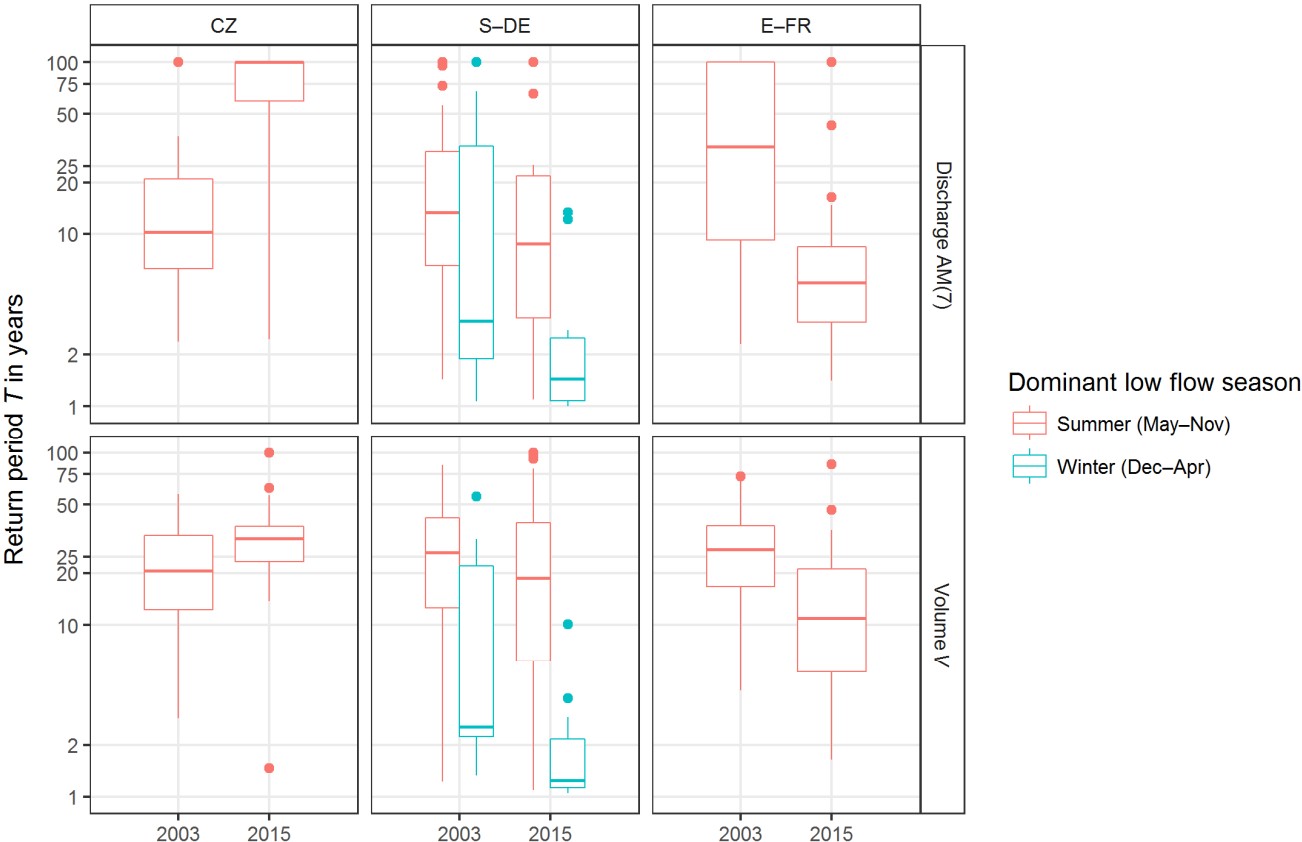

Figure 2. Regional distribution of return period $T$ (in yr) of low flow discharge AM(7) (upper panels) and deficit volume $V$ (lower panels) for the Czech Republic (left), S-Germany (centre) and E-France (right) (return periods > 100 yr not shown). For S-Germany, the blue boxplots represent alpine catchments with a winter low flow regime (mean day of occurrence of AM(7) between December and March). Boxes refer to upper quartile ($T_{75}$), median ($T_{med}$) and lower quartile ($T_{25}$) of the return period, dots represent the maximum range of outliers. Return period of about 2–10 years represents mild drought conditions, 10-50 years moderate conditions, 50-100 years severe conditions, and >100 years extreme conditions.

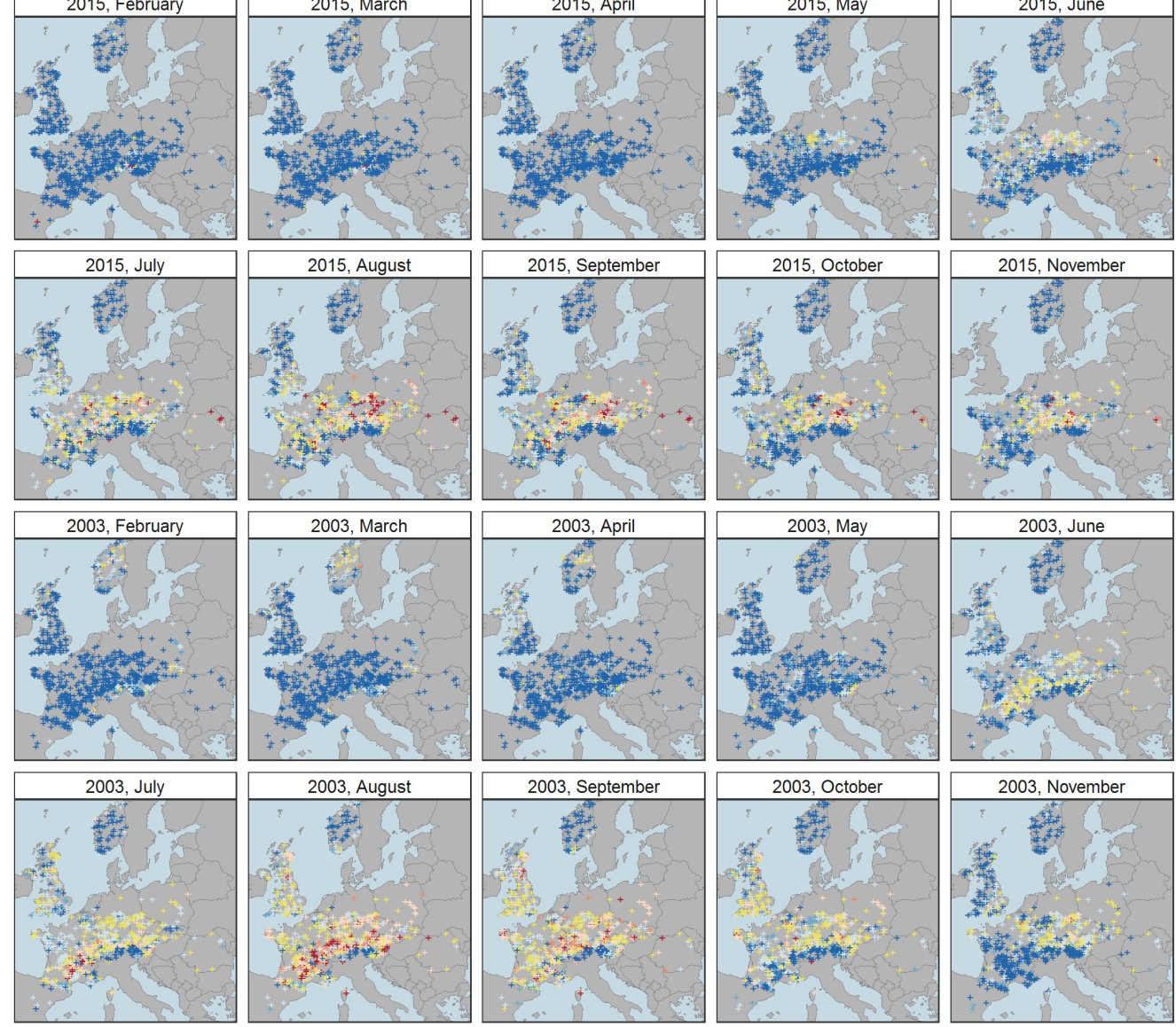

Figure 3. Return period T (in yr) of monthly 7-day minimum flows MM7. Colour codes are those of Fig. 1.

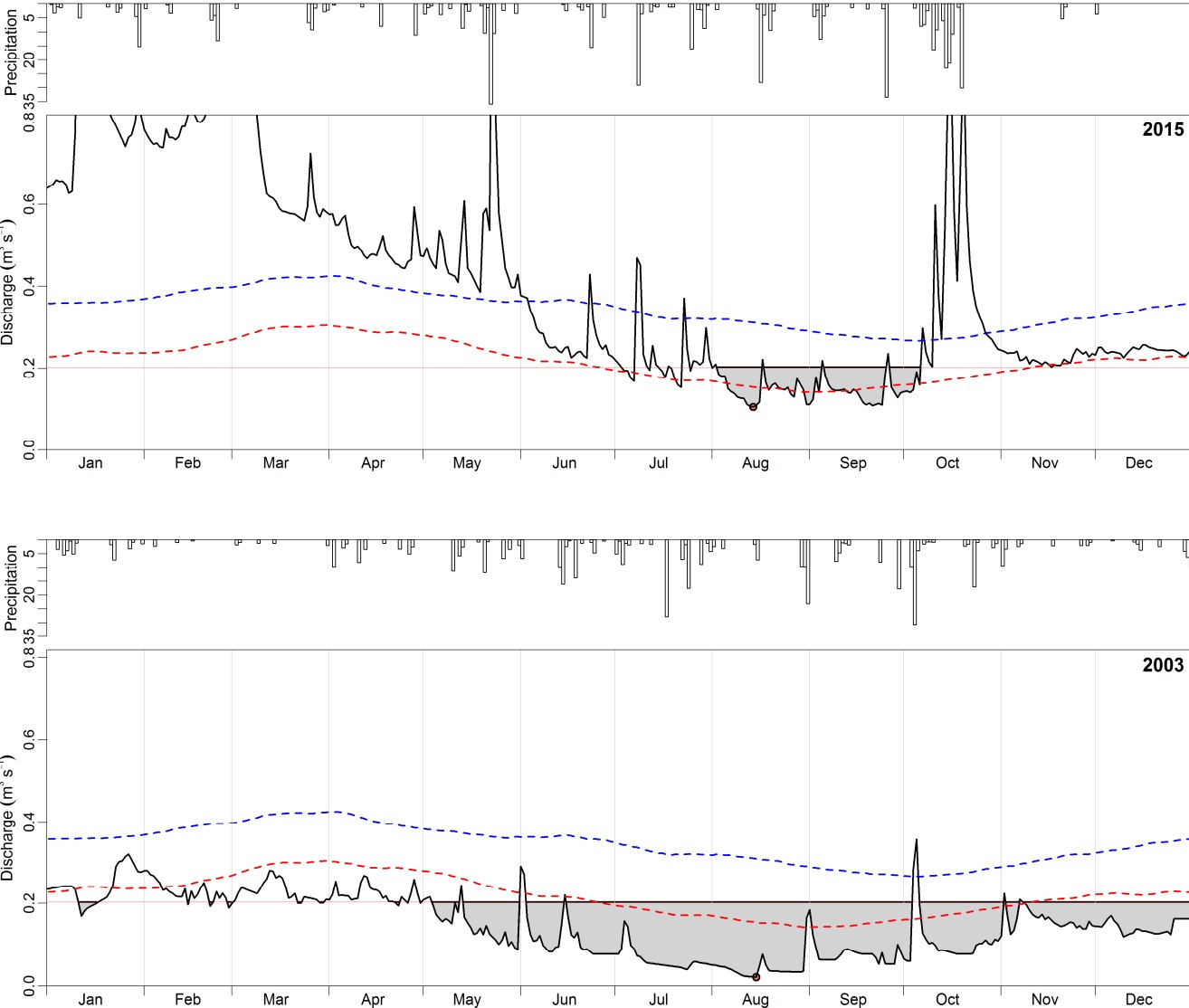

Figure 4. Hydrograph of gauge Altschlaining at river Tauchenbach in Austria (black line, large panels). Preconditions in 2015 (upper panel) were much wetter than in 2003 (lower panel). The grey polygon indicates the maximum annual low flow event below the annual threshold $Q_{80}$. The area of the polygon corresponds to the deficit volume, and its length (between onset and termination date) is the duration of the event. Dashed lines show seasonal varying thresholds $Q_{80s}$ (red) and $Q_{50s}$ (blue), corresponding to smoothed (30-day moving average) daily flow quantiles with exceedance probability 0.8 and 0.5. These lines are used to benchmark long-term average and dry seasonal conditions, respectively. Precipitation (daily sums in mm) shown in the smaller panels above the hydrographs.

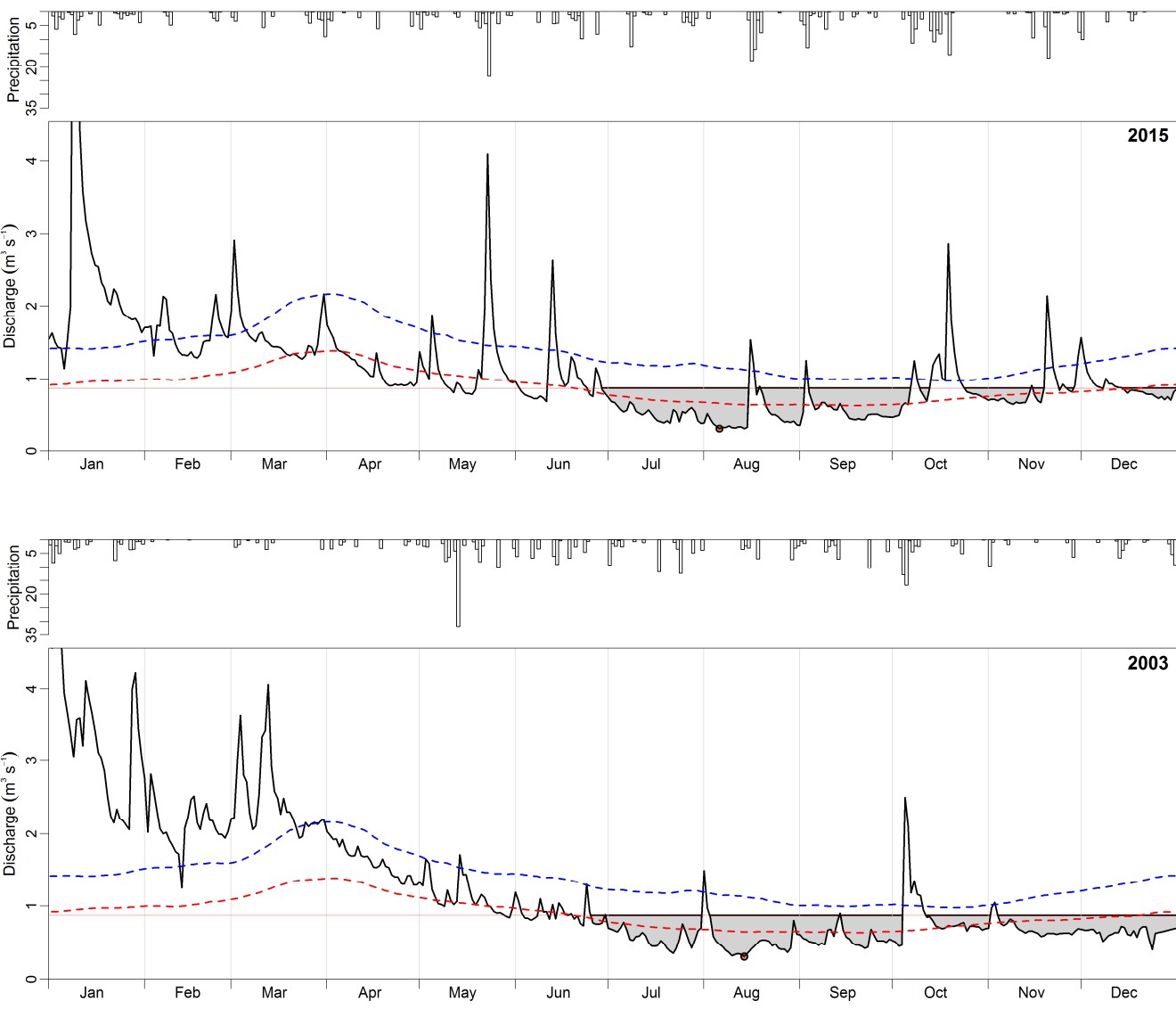

Figure 5. Hydrograph of gauge Imbach at river Krems, N-Austria (black line, large panels) together with weekly precipitation sums (mm, smaller panels above the hydrograph). Preconditions were much drier in 2015 than in 2003. Same signatures as Fig. 4.

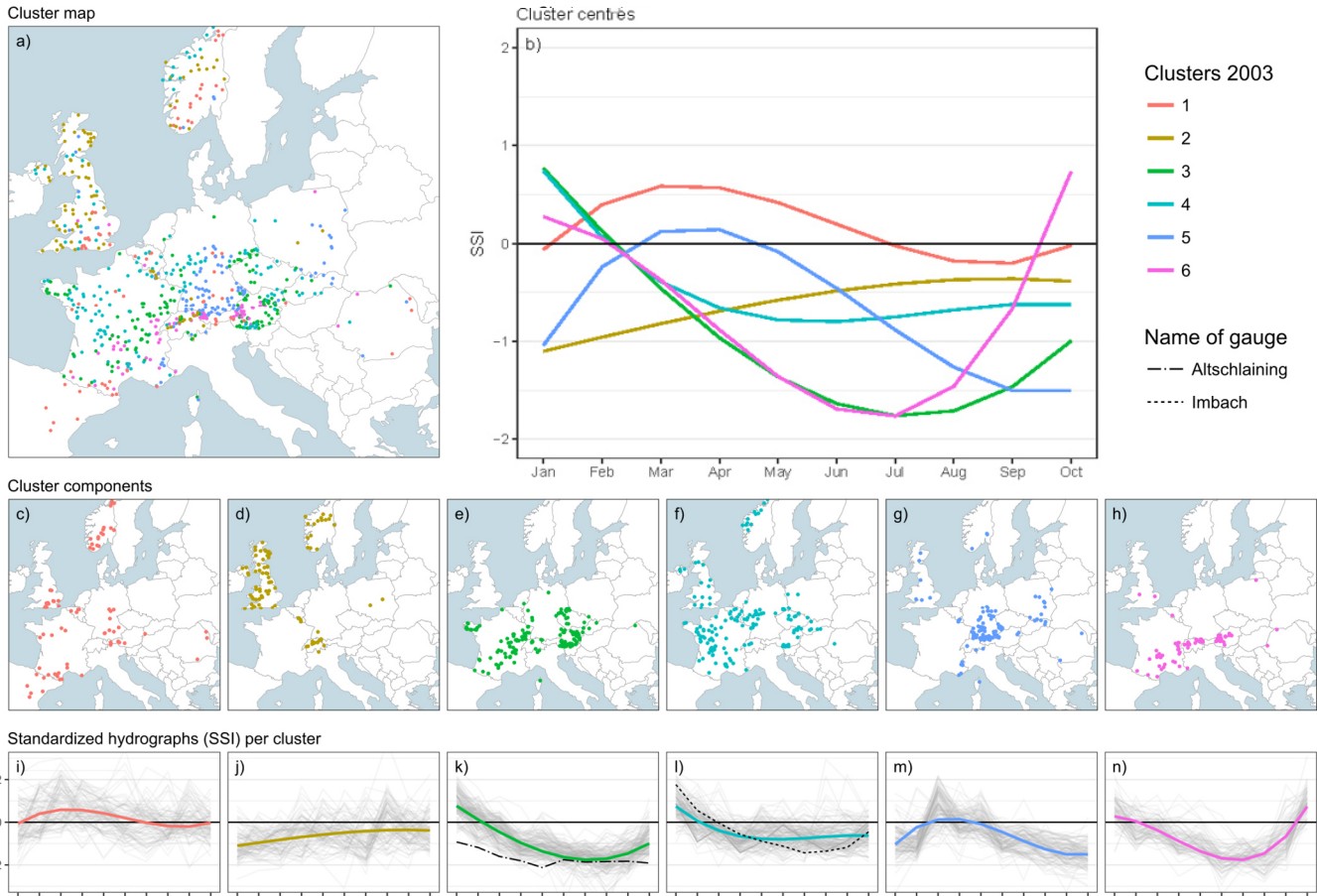

Figure 6. Clustering of the low flow event of 2003 based on monthly standardized streamflow index values SSI of the Jan –
Oct period. a) combined cluster map showing allocation of catchments to the clusters, b) combined map of functional models
of each cluster, c – h) cluster component maps, i – n) synoptic plots of standardized monthly hydrographs (indicated by thin
grey lines) of Cluster 1 – 6 together with the functional model of each cluster center (bold colored line). Altschlaining is
marked by a dotdashed black line, Imbach is marked by a dashed black line.

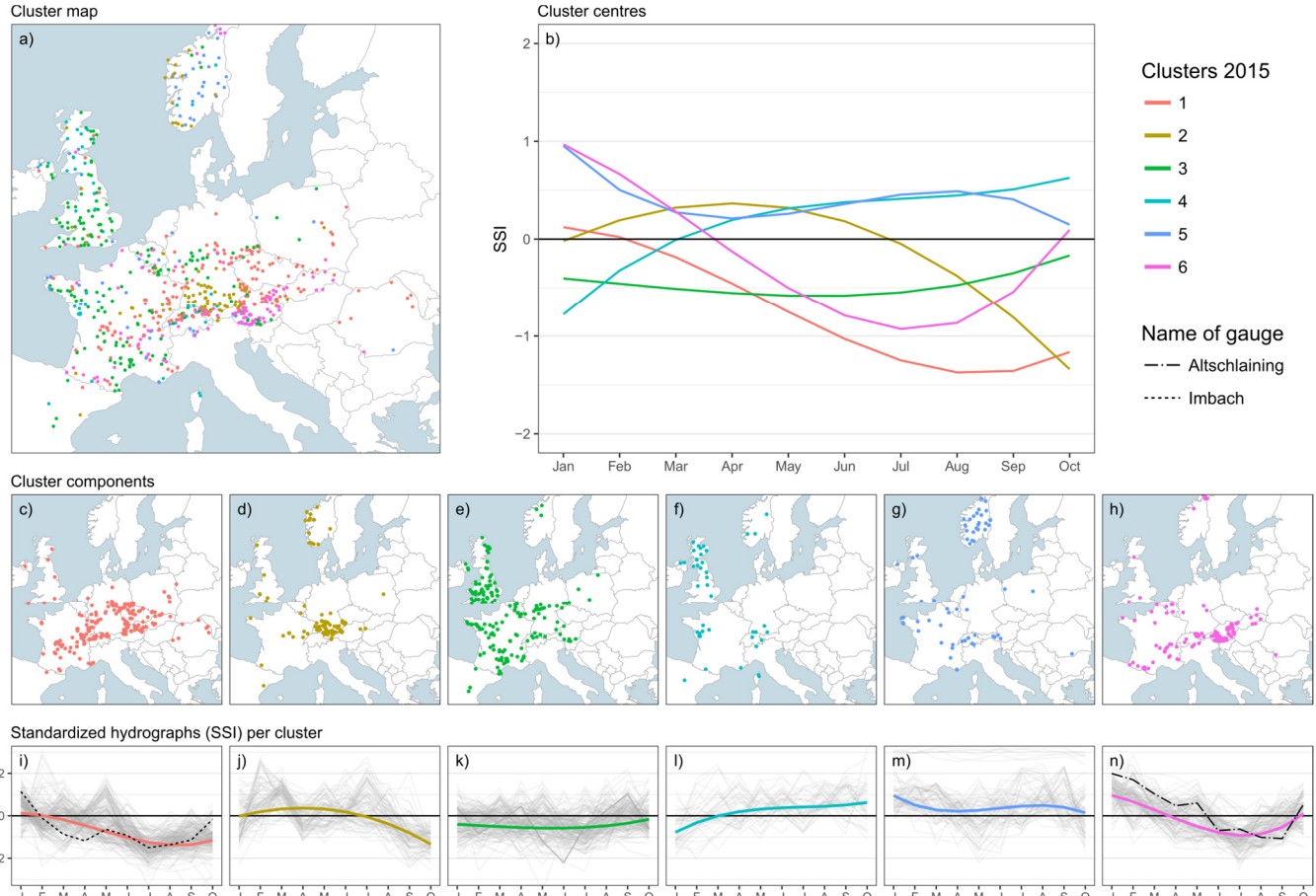

Figure 7. Clustering of the low flow event of 2015 based on monthly standardized streamflow index values SSI of the Jan – Oct period. a) combined cluster map showing allocation of catchments to the clusters, b) combined map of functional models of each cluster, c – h) cluster component maps, i – n) synoptic plots of standardized monthly hydrographs (indicated by thin grey lines) of Cluster 1 – 6 together with the functional model of each cluster center (bold colored line). Altschlaining is marked by a dotdashed black line, Imbach is marked by a dashed black line.

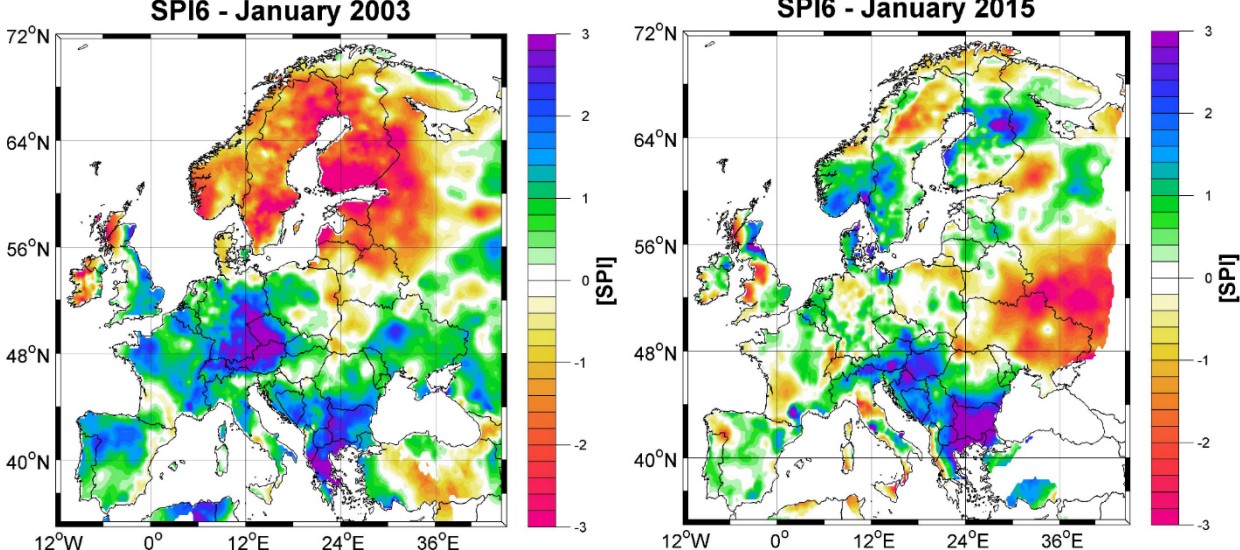

Figure 8. SPI6 for January 2003 (left) and January 2015 (right). Reference period 1971 – 2000.

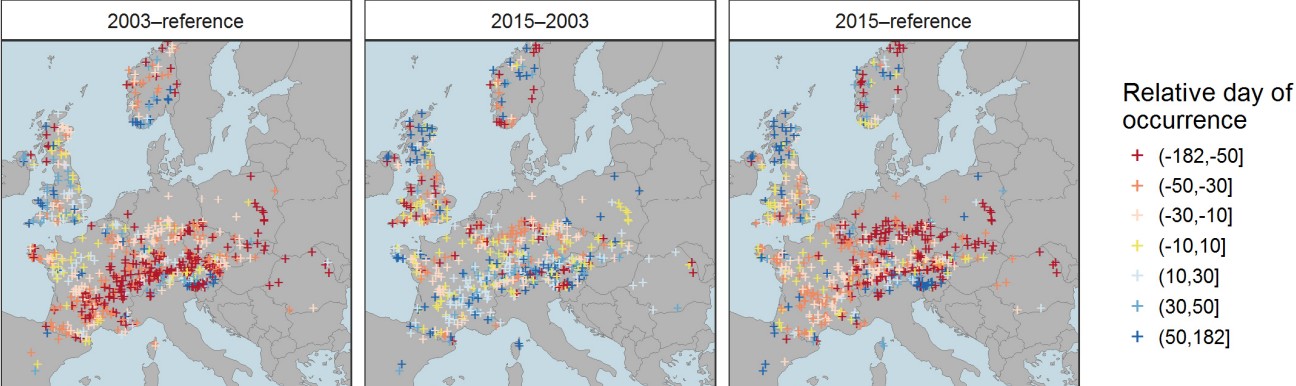

Figure 9. Relative day of occurrence $\Delta_\tau$ of the onset of the events. Left panels, of 2003 with respect to the reference period, central panels, of 2015 with respect to 2003, and right panels, of 2015 with respect to the reference period. Earlier occurrence (red) relate to relatively drier preconditions in winter or spring. Later occurrence (blue) relate to relatively wetter preconditions.

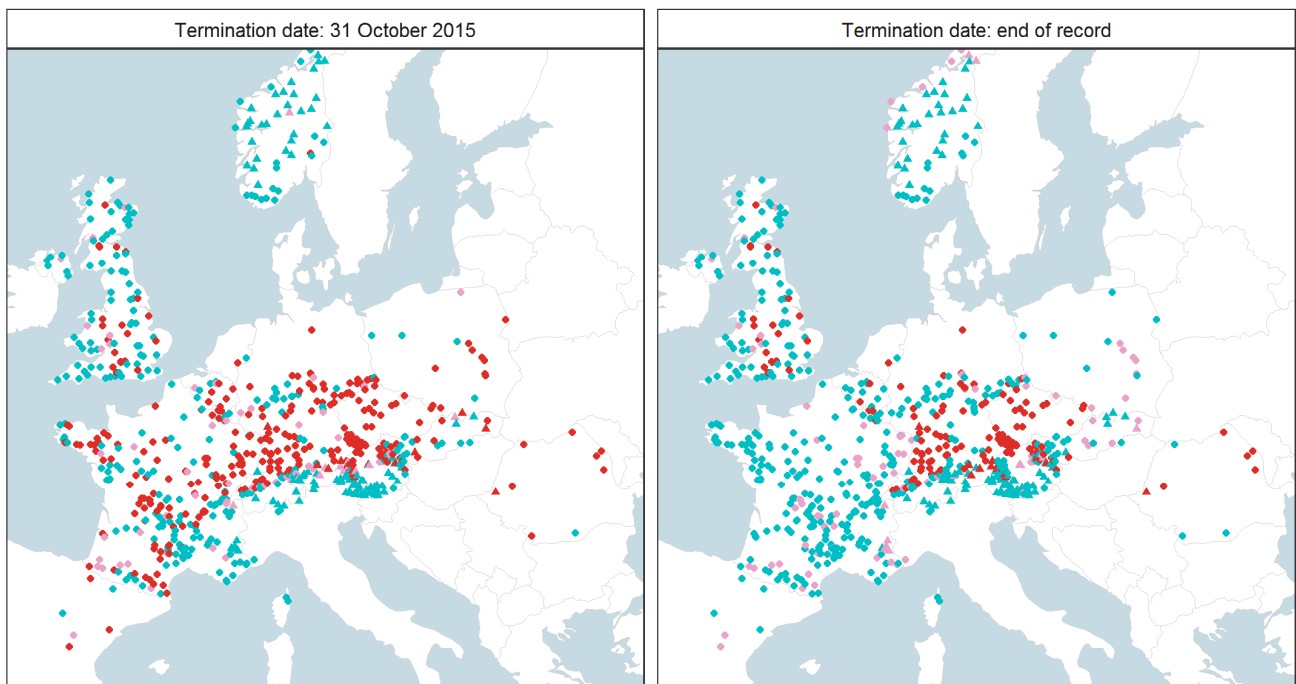

Figure 10. Stations potentially under drought at common termination date 31 October 2015 (left) and at end of record (10 November 2015 – 6 February 2016, variable between gauges). Red points indicate that the catchment has not totally recovered from the summer drought so that an event just after the end of record would be pooled by the SPA pooling.

Table A1: Statistical summary of low flow and streamflow drought characteristics of the event of 2015 (top) and 2003 (bottom) by country. min, q25, q50, q75, and max refer to sample quartiles of return periods (with non-exceedance probabilities of 0, 0.25, 0.50, 0.75 and 1). These statistics are also displayed in the boxplots (right panels, where red color refers to low flow discharge AM(7), green: duration $D$, blue: deficit volume $V$, and grey lines mark return periods T = 1, 25, 50, 75 and 100 yr). Countries abbreviated by ISO2 codes.

| | Discharge AM(7) | | | | | Duration $D$ | | | | | Volume $V$ | | | | | |
| | min | q25 | q50 | q75 | max | min | q25 | q50 | q75 | max | min | q25 | q50 | q75 | max | |
| **2015** | | | | | | | | | | | | | | | | |
| AT | 1.0 | 1.1 | 1.9 | 4.4 | 100.0 | 1.0 | 1.2 | 2.1 | 3.9 | 100.0 | 1.0 | 1.2 | 1.9 | 5.6 | 100.0 | |
| BE | 1.4 | 2.6 | 3.2 | 73.2 | 100.0 | 1.2 | 2.8 | 3.6 | 6.0 | 17.9 | 1.2 | 3.0 | 4.9 | 11.5 | 22.6 | |
| CZ | 2.5 | 59.3 | 100.0 | 100.0 | 100.0 | 1.3 | 10.0 | 12.6 | 16.2 | 24.2 | 1.5 | 23.3 | 31.9 | 37.5 | 100.0 | |
| FR | 1.0 | 1.6 | 2.7 | 4.6 | 100.0 | 1.1 | 1.7 | 3.1 | 7.8 | 56.2 | 1.1 | 1.6 | 3.2 | 8.1 | 85.6 | |
| DE | 1.0 | 2.8 | 7.9 | 18.0 | 100.0 | 1.0 | 3.0 | 6.1 | 10.9 | 29.3 | 1.1 | 4.2 | 7.8 | 21.7 | 100.0 | |
| NL | 2.3 | 2.3 | 2.3 | 2.6 | 2.8 | 3.4 | 3.4 | 3.4 | 5.4 | 7.4 | 1.6 | 2.1 | 2.7 | 5.1 | 7.6 | |
| NO | 1.0 | 1.0 | 1.1 | 1.4 | 27.8 | 1.0 | 1.1 | 1.1 | 1.5 | 6.0 | 1.0 | 1.1 | 1.1 | 1.3 | 8.3 | |
| PL | 1.2 | 1.6 | 4.2 | 11.7 | 39.1 | 1.2 | 2.9 | 4.7 | 7.5 | 36.3 | 1.1 | 3.4 | 10.5 | 14.9 | 22.1 | |
| RO | 1.7 | 7.0 | 9.6 | 100.0 | 100.0 | 1.2 | 2.4 | 6.8 | 12.1 | 100.0 | 1.2 | 4.7 | 18.3 | 26.3 | 100.0 | |
| SK | 1.1 | 2.0 | 2.7 | 5.1 | 22.0 | 1.1 | 1.5 | 1.9 | 4.8 | 82.2 | 1.1 | 1.9 | 2.3 | 4.2 | 47.5 | |
| ES | 1.3 | 1.4 | 1.9 | 2.4 | 100.0 | 1.2 | 1.2 | 1.2 | 1.4 | 3.5 | 1.2 | 1.2 | 1.3 | 1.8 | 4.5 | |
| CH | 1.0 | 1.1 | 2.3 | 4.7 | 100.0 | 1.1 | 1.5 | 4.0 | 8.2 | 65.2 | 1.0 | 1.3 | 2.5 | 9.3 | 100.0 | |
| GB | 1.0 | 1.2 | 1.6 | 2.3 | 9.7 | 1.0 | 1.2 | 1.6 | 2.5 | 17.1 | 1.0 | 1.2 | 1.5 | 2.2 | 8.7 | |
| **2003** | | | | | | | | | | | | | | | | |
| AT | 1.0 | 1.9 | 4.1 | 13.5 | 100.0 | 1.1 | 1.8 | 3.5 | 17.5 | 100.0 | 1.1 | 1.9 | 4.0 | 14.9 | 100.0 | |
| BE | 1.2 | 2.0 | 2.9 | 5.3 | 100.0 | 1.1 | 2.5 | 6.4 | 13.4 | 30.4 | 1.2 | 2.1 | 5.2 | 16.9 | 46.0 | |
| CZ | 2.4 | 6.3 | 10.2 | 21.0 | 100.0 | 4.7 | 8.8 | 14.4 | 29.5 | 34.2 | 2.9 | 12.3 | 20.6 | 33.0 | 57.4 | |
| FR | 1.2 | 2.7 | 5.7 | 22.6 | 100.0 | 1.1 | 3.5 | 6.9 | 12.1 | 68.7 | 1.1 | 3.5 | 9.1 | 20.2 | 87.5 | |
| DE | 1.1 | 3.9 | 8.4 | 26.7 | 100.0 | 1.2 | 6.6 | 13.3 | 24.1 | 69.3 | 1.2 | 7.7 | 15.6 | 33.5 | 85.4 | |
| NL | 1.0 | 8.6 | 16.2 | 17.2 | 18.3 | 4.2 | 7.8 | 11.3 | 14.9 | 18.4 | 7.2 | 10.4 | 13.5 | 16.7 | 19.8 | |
| NO | 1.0 | 1.4 | 2.1 | 3.4 | 100.0 | 1.0 | 1.4 | 2.2 | 4.4 | 97.7 | 1.1 | 1.7 | 2.3 | 4.0 | 41.9 | |
| PL | 3.3 | 7.0 | 10.2 | 14.0 | 100.0 | 3.4 | 6.0 | 13.3 | 21.5 | 39.8 | 3.1 | 7.0 | 16.9 | 35.5 | 46.2 | |
| RO | 1.7 | 2.9 | 4.1 | 15.3 | 100.0 | 1.5 | 3.5 | 5.0 | 13.4 | 31.5 | 1.4 | 3.3 | 4.8 | 16.1 | 62.1 | |
| SK | 2.0 | 4.1 | 7.4 | 20.0 | 100.0 | 3.4 | 15.6 | 21.8 | 29.0 | 60.4 | 4.5 | 9.6 | 18.6 | 40.3 | 54.9 | |
| ES | 1.0 | 1.6 | 1.8 | 1.9 | 2.6 | 1.1 | 1.2 | 1.6 | 1.7 | 3.9 | 1.1 | 1.4 | 1.4 | 1.5 | 6.4 | |
| CH | 1.0 | 2.8 | 9.2 | 49.9 | 100.0 | 1.0 | 2.3 | 5.2 | 14.1 | 100.0 | 1.1 | 3.0 | 10.3 | 23.4 | 100.0 | |
| GB | 1.2 | 2.2 | 3.1 | 5.9 | 100.0 | 1.2 | 2.6 | 4.6 | 11.6 | 79.8 | 1.2 | 2.7 | 5.2 | 11.5 | 85.1 | |

Table A2: Statistical summary of low flow and streamflow drought characteristics of stations with predominant summer seasonality: 2015 (top) and 2003 (bottom). Symbols and legend are those of Table A1.

| | Discharge AM(7) | | | | | Duration D | | | | | Volume V | | | | | |
|---|---|---|---|---|---|---|---|---|---|---|---|---|---|---|---|---|
| | min | q25 | q50 | q75 | max | min | q25 | q50 | q75 | max | min | q25 | q50 | q75 | max | |
| **2015** | | | | | | | | | | | | | | | | |
| AT | 1.0 | 2.2 | 3.5 | 11.1 | 100.0 | 1.1 | 2.4 | 3.8 | 6.3 | 12.6 | 1.1 | 2.1 | 4.7 | 10.0 | 51.8 | |
| BE | 1.4 | 2.6 | 3.2 | 73.2 | 100.0 | 1.2 | 2.8 | 3.6 | 6.0 | 17.9 | 1.2 | 3.0 | 4.9 | 11.5 | 22.6 | |
| CZ | 2.5 | 59.3 | 100.0 | 100.0 | 100.0 | 1.3 | 10.0 | 12.6 | 16.2 | 24.2 | 1.5 | 23.3 | 31.9 | 37.5 | 100.0 | |
| FR | 1.0 | 1.7 | 2.7 | 4.8 | 100.0 | 1.1 | 1.8 | 3.3 | 7.9 | 56.2 | 1.1 | 1.6 | 3.2 | 8.3 | 85.6 | |
| DE | 1.1 | 3.5 | 8.7 | 21.2 | 100.0 | 1.1 | 3.6 | 6.7 | 11.2 | 29.3 | 1.1 | 5.4 | 8.3 | 23.8 | 100.0 | |
| NL | 2.3 | 2.3 | 2.3 | 2.6 | 2.8 | 3.4 | 3.4 | 3.4 | 5.4 | 7.4 | 1.6 | 2.1 | 2.7 | 5.1 | 7.6 | |
| NO | 1.0 | 1.1 | 1.1 | 1.3 | 3.1 | 1.1 | 1.1 | 1.1 | 1.6 | 2.3 | 1.0 | 1.1 | 1.1 | 1.4 | 3.2 | |
| PL | 1.2 | 1.7 | 4.4 | 12.6 | 39.1 | 1.2 | 2.5 | 4.6 | 6.6 | 36.3 | 1.1 | 3.8 | 10.7 | 15.2 | 22.1 | |
| RO | 1.7 | 8.5 | 39.3 | 100.0 | 100.0 | 1.2 | 3.9 | 8.8 | 14.4 | 100.0 | 1.2 | 8.1 | 21.3 | 36.9 | 100.0 | |
| SK | 1.2 | 2.4 | 3.6 | 5.1 | 22.0 | 1.1 | 1.5 | 1.9 | 6.5 | 82.2 | 1.1 | 2.0 | 2.4 | 5.0 | 47.5 | |
| ES | 1.3 | 1.4 | 1.9 | 2.4 | 100.0 | 1.2 | 1.2 | 1.2 | 1.4 | 3.5 | 1.2 | 1.2 | 1.3 | 1.8 | 4.5 | |
| CH | 1.0 | 1.6 | 3.3 | 5.6 | 100.0 | 1.3 | 1.9 | 5.2 | 11.5 | 65.2 | 1.3 | 2.1 | 4.2 | 11.4 | 100.0 | |
| GB | 1.0 | 1.2 | 1.6 | 2.3 | 9.7 | 1.0 | 1.2 | 1.6 | 2.5 | 17.1 | 1.0 | 1.2 | 1.5 | 2.2 | 8.7 | |
| **2003** | | | | | | | | | | | | | | | | |
| AT | 1.2 | 4.8 | 12.3 | 26.6 | 100.0 | 1.1 | 6.7 | 15.2 | 29.0 | 92.7 | 1.1 | 6.3 | 15.7 | 23.8 | 100.0 | |
| BE | 1.2 | 2.0 | 2.9 | 5.3 | 100.0 | 1.1 | 2.5 | 6.4 | 13.4 | 30.4 | 1.2 | 2.1 | 5.2 | 16.9 | 46.0 | |
| CZ | 2.4 | 6.3 | 10.2 | 21.0 | 100.0 | 4.7 | 8.8 | 14.4 | 29.5 | 34.2 | 2.9 | 12.3 | 20.6 | 33.0 | 57.4 | |
| FR | 1.2 | 2.9 | 5.7 | 22.8 | 100.0 | 1.1 | 3.6 | 7.2 | 12.2 | 68.7 | 1.1 | 3.8 | 9.4 | 20.3 | 87.5 | |
| DE | 1.2 | 4.3 | 9.1 | 26.5 | 100.0 | 1.2 | 7.1 | 15.2 | 24.9 | 69.3 | 1.2 | 7.8 | 17.3 | 34.4 | 85.4 | |
| NL | 1.0 | 8.6 | 16.2 | 17.2 | 18.3 | 4.2 | 7.8 | 11.3 | 14.9 | 18.4 | 7.2 | 10.4 | 13.5 | 16.7 | 19.8 | |
| NO | 1.1 | 1.3 | 1.7 | 2.5 | 8.1 | 1.1 | 1.4 | 1.9 | 2.7 | 67.8 | 1.1 | 1.3 | 1.8 | 2.3 | 6.3 | |
| PL | 3.4 | 7.2 | 10.4 | 16.1 | 100.0 | 3.4 | 7.0 | 17.7 | 23.0 | 39.8 | 3.1 | 8.6 | 18.0 | 36.1 | 46.2 | |
| RO | 1.7 | 2.7 | 5.1 | 16.9 | 100.0 | 1.5 | 3.5 | 7.7 | 16.5 | 31.5 | 1.4 | 4.0 | 10.4 | 16.1 | 62.1 | |
| SK | 2.0 | 4.6 | 9.2 | 27.7 | 100.0 | 3.4 | 12.6 | 22.3 | 29.0 | 60.4 | 4.5 | 9.0 | 22.7 | 40.3 | 54.9 | |
| ES | 1.0 | 1.6 | 1.8 | 1.9 | 2.6 | 1.1 | 1.2 | 1.6 | 1.7 | 3.9 | 1.1 | 1.4 | 1.4 | 1.5 | 6.4 | |
| CH | 1.3 | 5.0 | 12.9 | 56.6 | 100.0 | 1.9 | 2.6 | 8.9 | 16.0 | 100.0 | 1.4 | 4.0 | 12.4 | 26.8 | 100.0 | |
| GB | 1.2 | 2.2 | 3.1 | 5.9 | 100.0 | 1.2 | 2.6 | 4.6 | 11.6 | 79.8 | 1.2 | 2.7 | 5.2 | 11.5 | 85.1 | |

Table A3: Statistical summary of low flow and streamflow drought characteristics of stations with predominant winter seasonality: 2015 (top) and 2003 (bottom). Symbols and legend are those of Table A1.

| | Discharge AM(7) | | | | | Duration $D$ | | | | | Volume $V$ | | | | | |
|---|---|---|---|---|---|---|---|---|---|---|---|---|---|---|---|---|
| | min | q25 | q50 | q75 | max | min | q25 | q50 | q75 | max | min | q25 | q50 | q75 | max | |
| **2015** | | | | | | | | | | | | | | | | |
| AT | 1.0 | 1.0 | 1.2 | 2.1 | 100.0 | 1.0 | 1.1 | 1.3 | 2.2 | 100.0 | 1.0 | 1.1 | 1.2 | 1.9 | 100.0 | |
| FR | 1.1 | 1.1 | 1.5 | 1.9 | 2.0 | 1.1 | 1.3 | 1.5 | 2.2 | 2.9 | 1.1 | 1.3 | 1.4 | 1.8 | 2.2 | |
| DE | 1.0 | 1.1 | 1.4 | 2.5 | 13.5 | 1.0 | 1.1 | 1.5 | 1.6 | 3.9 | 1.1 | 1.1 | 1.2 | 2.2 | 10.1 | |
| NO | 1.0 | 1.0 | 1.1 | 1.4 | 27.8 | 1.0 | 1.1 | 1.1 | 1.5 | 6.0 | 1.0 | 1.1 | 1.1 | 1.2 | 8.3 | |
| PL | 1.4 | 1.4 | 1.4 | 1.4 | 1.4 | 11.1 | 11.1 | 11.1 | 11.1 | 11.1 | 3.1 | 3.1 | 3.1 | 3.1 | 3.1 | |
| RO | 2.1 | 2.1 | 2.1 | 2.1 | 2.1 | 2.3 | 2.3 | 2.3 | 2.3 | 2.3 | 2.3 | 2.3 | 2.3 | 2.3 | 2.3 | |
| SK | 1.1 | 1.8 | 2.2 | 5.5 | 14.5 | 1.1 | 1.6 | 2.0 | 2.8 | 4.2 | 1.1 | 1.8 | 2.1 | 3.5 | 7.4 | |
| CH | 1.0 | 1.0 | 1.1 | 1.1 | 3.2 | 1.1 | 1.1 | 1.1 | 1.2 | 6.4 | 1.0 | 1.1 | 1.1 | 1.1 | 2.2 | |
| **2003** | | | | | | | | | | | | | | | | |
| AT | 1.0 | 1.4 | 2.3 | 4.4 | 100.0 | 1.1 | 1.4 | 2.0 | 3.6 | 100.0 | 1.1 | 1.5 | 2.1 | 3.8 | 100.0 | |
| FR | 1.3 | 1.5 | 2.2 | 7.5 | 28.3 | 1.2 | 1.2 | 1.5 | 1.6 | 3.0 | 1.2 | 1.3 | 1.8 | 2.2 | 3.0 | |
| DE | 1.1 | 1.9 | 3.1 | 32.5 | 100.0 | 1.6 | 1.9 | 3.2 | 11.8 | 47.0 | 1.3 | 2.2 | 2.5 | 22.1 | 55.6 | |
| NO | 1.0 | 1.6 | 2.2 | 3.6 | 100.0 | 1.0 | 1.6 | 2.6 | 5.7 | 97.7 | 1.1 | 1.8 | 2.8 | 4.5 | 41.9 | |
| PL | 3.3 | 3.3 | 3.3 | 3.3 | 3.3 | 3.5 | 3.5 | 3.5 | 3.5 | 3.5 | 5.8 | 5.8 | 5.8 | 5.8 | 5.8 | |
| RO | 4.1 | 4.1 | 4.1 | 4.1 | 4.1 | 4.3 | 4.3 | 4.3 | 4.3 | 4.3 | 3.3 | 3.3 | 3.3 | 3.3 | 3.3 | |
| SK | 2.8 | 3.6 | 4.9 | 8.9 | 18.0 | 15.7 | 16.0 | 19.5 | 27.4 | 40.6 | 8.6 | 10.8 | 16.1 | 27.2 | 46.9 | |
| CH | 1.0 | 1.2 | 1.4 | 1.8 | 4.9 | 1.0 | 1.1 | 1.2 | 2.9 | 4.6 | 1.1 | 1.1 | 1.3 | 2.6 | 23.4 | |

