# Peer review of "The European 2015 drought from a hydrological perspective"

_Hydrology and Earth System Sciences, 2016_

## Referee Comment (RC1) · Anonymous Referee #1 · 23 Sep 2016

General:

The paper is well written; it is highly descriptive and provides relevant facts on impacts of droughts in different countries for the year 2015. However for the reader it would have been easier to if the different aspects of droughts would have been summarized in a table for the different countries. One aspect however is missing when talking about droughts for a year. It is very important to provide information about the meteorological conditions of the previous year, especially if you look at the status of the aquifers.

Conclusions:

A European network experts on water scarcity and droughts produced a report in 2007 on drought management plans as part of the Common Implementation Strategy of the Water Framework Directive, which was endorsed by the member States in November

2007. The report sets out recommendations in preparing drought management plans. Examples are shown for some Member States. The authors could have explored in the first place the status of the implementation in the Member States instead of referring to the example given by GWP and WMO.

―――――――――――――――――――

---

## Referee Comment (RC2) · Anonymous Referee #2 · 4 Nov 2016

The manuscript describes the 2015 streamflow drought event relative to the 2003 event based on observed low flow conditions derived over a set of stations. Results are compared also with the corresponding 2015 meteorological drought event analysed in detail in a companion work. I find that the analysis presented provides limited advances for a better understanding of hydrological drought processes and many parts of the manuscript are too qualitative and descriptive. My major concerns are: 1) The influence of antecedent moisture conditions on drought developments is interesting and novel aspect, maybe the most relevant in the work. However it is analysed only on two stations. I suggest the authors to extent this investigation on the whole set of data to derive their conclusions in a more robust manner and spatially over the domain. The use of cluster analysis (or similar more objective techniques) to group stations with similar hydro-meteorological response may be an option. More details on the characteristics of antecedent moisture conditions, for instance timing and magnitude of antecedent precipitation that may reduce the probability of subsequent extreme drought events, would be relevant as well in view of an enhanced predictability and monitoring of low flow conditions. 2) Section 5.2 is too descriptive and qualitative, and it does not add relevant new knowledge. Furthermore, it suffers of a poor methodological approach. There have been developed automated research algorithms to collect events and information from web and media in a systematic manner. I suggest the authors to implement such methods or to remove completely this section. 3) Description of methods needs to be improved. In particular it is not specified what time series is used to derive fitting functions and return periods. I suppose the reference period, but this should be better clarified.

In the comparison with meteorological droughts the following references may be relevant. Bachmair et al., 2016 (http://onlinelibrary.wiley.com/doi/10.1002/wat2.1154/full) an Loon and Laaha, 2015 (http://www.sciencedirect.com/science/article/pii/S0022169414008543). Barker et al., 2016 (http://www.hydrol-earth-syst-sci.net/20/2483/2016/),

Minor comments: Page 1, line 29: please, consider to remove "in this second paper" Page 1, line 30: stream gauge stations instead of records? Page 2, line 11: please, add the relevant references to support this sentence Page 2, line 16: "Droughts... to analyse", too vague, consider to rephrase or remove. Page 2, line 19-20: This concept needs to be better expressed. Page 3, line 1: move the reference to the end of the sentence. Page 3, line 8: please do not abbreviate South, North, East and West throughout the manuscript. Such abbreviation is not a standard. Page 5 lines 9-13: this information is not relevant for this work, consider removing it. Page 5, line 15: the 2013 is also compared to the 2003 event, this should be clarified. Page 5 lines 20-25: I would suggest to synthesize this content and move to the next sections for a better organization of the text and to avoid redundancy. Are the low-flow indices calculated for the 2015, 2003 and reference period? Please, clarify. Page 6, line 18: how do you

define "totally recovered", please clarify. Page 6, Section 3.2. Please clarify on what time series you estimate the fitting functions to derive return periods for reference period, 2015 and 2003. Why did you use such fitting functions instead of generalized extreme value or pareto distribution? Page 7 line 21: contrasting response instead of dipole? Page 7, line 23: the patter discussed seems not including North-Austria. Maybe, because the graphical representation is not very clear in colours and symbols. I strongly suggest improving figures with maps by showing more contrasted colours. Page 7, line 26: the 2015 drought-affected area? Clarify Page 8, line 10: Please, clarify. Page 9, line 3-4: Consider to remove winter plots from the graph, they do not add relevant information. Page 9, line 13: low flow threshold? I understood that you were looking at the minimum flows here. Please clarify. Page 10 Section 4.5 please, rephrase without using bullet points. Page 14 line 7: the extreme is most extreme... please rephrase. Page 14 line 25: add "streamflow" to drought. Page 15, line 10-11. Is there any additional drought self-propagation mechanism linked to land-atmosphere interactions that could contribute in explaining these processes? Dry soils may lead to lower probability of precipitation and thus cause intensified droughts. See for instance Senevitarne et al. 2010 (Earth-Science Reviews 99 (2010) 125–161). Page 15 lines 20-33. This text is very speculative and not related to the work presented, please consider removing it.

Please also note the supplement to this comment:
http://www.hydrol-earth-syst-sci-discuss.net/hess-2016-366/hess-2016-366-RC2-supplement.pdf

---

## Author Comment (AC1) · 16 Dec 2016

**Response to the comment of Anonymous Referee # 1**

*We would like to thank the reviewer for his kind assessment of the manuscript. Below is our response to the issues raised in the review. The original comment is printed in plain font, our response is printed in italics.*

General:
The paper is well written; it is highly descriptive and provides relevant facts on impacts of droughts in different countries for the year 2015. However for the reader it would have been easier to if the different aspects of droughts would have been summarized in a table for the different countries. One aspect however is missing when talking about droughts for a year. It is very important to provide information about the meteorological conditions of the previous year, especially if you look at the status of the aquifers.

*We thank the referee for this excellent evaluation.*
*We suggest that we will provide a table that summarizes the different aspects of drought by country. A draft table is contained in the Appendix (Tab. S1)*
*We further suggest to present the January-SPEI(6) (of Aug—Jan) (Table S1) (whose time scale has been optimized for covering not only fall and winter precipitation but also the major precipitation events of August 2002) to summarize antecedent conditions. A draft of the Figure is shown in Fig. S1 illustrating that the mayor rainfall event is well represented by the figure.*

Conclusions:
A European network experts on water scarcity and droughts produced a report in 2007 on drought management plans as part of the Common Implementation Strategy of the Water Framework Directive, which was endorsed by the member States in November 2007. The report sets out recommendations in preparing drought management plans. Examples are shown for some Member States. The authors could have explored in the first place the status of the implementation in the Member States instead of referring to the example given by GWP and WMO.

*We appreciate the reminder and suggest that we will refer to the report and the need for hydrological flow indices in the introduction and again pick it up in the discussion. The status of RBMP implementation appears not directly relevant to this study's objective of understanding the physical hydrological aspects of extent and propagation of the hydrological drought, but we agree that the report helps to highlight the need and potential benefits of common indices.*

*Reference:*
*Water Scarcity and Droughts Expert Network: Drought Management Plan Report - Including Agricultural, Drought Indicators and Climate Change Aspects. [online] Available from: http://ec.europa.eu/environment/water/quantity/pdf/dmp_report.pdf (Accessed 30 November 2016), 2007.*

**Figures**

[Figure]

*Figure S1. SPI6 for January 2003 (left) and January 2015 (right). Reference period 1971-2000.*

**Tables**

Table S1: Statistical summary of low flow and streamflow drought characteristics of 2003 (top) and 2015 (bottom) by country.

| | Discharge | | | | | Duration | | | | | Volume | | | | | |
|------|------|------|------|------|-------|------|------|------|------|-------|------|------|------|------|-------|---|
| | min | q25 | q50 | q75 | max | min | q25 | q50 | q75 | max | min | q25 | q50 | q75 | max | |
| **2003** | | | | | | | | | | | | | | | | |
| at | 1.0 | 1.9 | 4.1 | 13.5 | 100.0 | 1.1 | 1.8 | 3.5 | 17.5 | 100.0 | 1.1 | 1.9 | 4.0 | 14.9 | 100.0 | |
| be | 1.2 | 2.0 | 2.9 | 5.3 | 100.0 | 1.1 | 2.5 | 6.4 | 13.4 | 30.4 | 1.2 | 2.1 | 5.2 | 16.9 | 46.0 | |
| cz | 2.4 | 6.3 | 10.2 | 21.0 | 100.0 | 4.7 | 8.8 | 14.4 | 29.5 | 34.2 | 2.9 | 12.3 | 20.6 | 33.0 | 57.4 | |
| fr | 1.2 | 2.7 | 5.7 | 22.6 | 100.0 | 1.1 | 3.5 | 6.9 | 12.1 | 68.7 | 1.1 | 3.5 | 9.1 | 20.2 | 87.5 | |
| de | 1.1 | 3.9 | 8.4 | 26.7 | 100.0 | 1.2 | 6.6 | 13.3 | 24.1 | 69.3 | 1.2 | 7.7 | 15.6 | 33.5 | 85.4 | |
| nl | 1.0 | 8.6 | 16.2 | 17.2 | 18.3 | 4.2 | 7.8 | 11.3 | 14.9 | 18.4 | 7.2 | 10.4 | 13.5 | 16.7 | 19.8 | |
| no | 1.0 | 1.4 | 2.1 | 3.4 | 100.0 | 1.0 | 1.4 | 2.2 | 4.4 | 97.7 | 1.1 | 1.7 | 2.3 | 4.0 | 41.9 | |
| pl | 3.3 | 7.0 | 10.2 | 14.0 | 100.0 | 3.4 | 6.0 | 13.3 | 21.5 | 39.8 | 3.1 | 7.0 | 16.9 | 35.5 | 46.2 | |
| ro | 1.7 | 2.9 | 4.1 | 15.3 | 100.0 | 1.5 | 3.5 | 5.0 | 13.4 | 31.5 | 1.4 | 3.3 | 4.8 | 16.1 | 62.1 | |
| sk | 2.0 | 4.1 | 7.4 | 20.0 | 100.0 | 3.4 | 15.6 | 21.8 | 29.0 | 60.4 | 4.5 | 9.6 | 18.6 | 40.3 | 54.9 | |
| es | 1.0 | 1.6 | 1.8 | 1.9 | 2.6 | 1.1 | 1.2 | 1.6 | 1.7 | 3.9 | 1.1 | 1.4 | 1.4 | 1.5 | 6.4 | |
| ch | 1.0 | 2.8 | 9.2 | 49.9 | 100.0 | 1.0 | 2.3 | 5.2 | 14.1 | 100.0 | 1.1 | 3.0 | 10.3 | 23.4 | 100.0 | |
| gb | 1.2 | 2.2 | 3.1 | 5.9 | 100.0 | 1.2 | 2.6 | 4.6 | 11.6 | 79.8 | 1.2 | 2.7 | 5.2 | 11.5 | 85.1 | |
| **2015** | | | | | | | | | | | | | | | | |
| at | 1.0 | 1.1 | 1.9 | 4.4 | 100.0 | 1.0 | 1.2 | 2.1 | 3.9 | 100.0 | 1.0 | 1.2 | 1.9 | 5.6 | 100.0 | |
| be | 1.4 | 2.6 | 3.2 | 73.2 | 100.0 | 1.2 | 2.8 | 3.6 | 6.0 | 17.9 | 1.2 | 3.0 | 4.9 | 11.5 | 22.6 | |
| cz | 2.5 | 59.3 | 100.0 | 100.0 | 100.0 | 1.3 | 10.0 | 12.6 | 16.2 | 24.2 | 1.5 | 23.3 | 31.9 | 37.5 | 100.0 | |
| fr | 1.0 | 1.6 | 2.7 | 4.6 | 100.0 | 1.1 | 1.7 | 3.1 | 7.8 | 56.2 | 1.1 | 1.6 | 3.2 | 8.1 | 85.6 | |
| de | 1.0 | 2.8 | 7.9 | 18.0 | 100.0 | 1.0 | 3.0 | 6.1 | 10.9 | 29.3 | 1.1 | 4.2 | 7.8 | 21.7 | 100.0 | |
| nl | 2.3 | 2.3 | 2.3 | 2.6 | 2.8 | 3.4 | 3.4 | 3.4 | 5.4 | 7.4 | 1.6 | 2.1 | 2.7 | 5.1 | 7.6 | |
| no | 1.0 | 1.0 | 1.1 | 1.4 | 27.8 | 1.0 | 1.1 | 1.1 | 1.5 | 6.0 | 1.0 | 1.1 | 1.1 | 1.3 | 8.3 | |
| pl | 1.2 | 1.6 | 4.2 | 11.7 | 39.1 | 1.2 | 2.9 | 4.7 | 7.5 | 36.3 | 1.1 | 3.4 | 10.5 | 14.9 | 22.1 | |
| ro | 1.7 | 7.0 | 9.6 | 100.0 | 100.0 | 1.2 | 2.4 | 6.8 | 12.1 | 100.0 | 1.2 | 4.7 | 18.3 | 26.3 | 100.0 | |
| sk | 1.1 | 2.0 | 2.7 | 5.1 | 22.0 | 1.1 | 1.5 | 1.9 | 4.8 | 82.2 | 1.1 | 1.9 | 2.3 | 4.2 | 47.5 | |
| es | 1.3 | 1.4 | 1.9 | 2.4 | 100.0 | 1.2 | 1.2 | 1.2 | 1.4 | 3.5 | 1.2 | 1.2 | 1.3 | 1.8 | 4.5 | |
| ch | 1.0 | 1.1 | 2.3 | 4.7 | 100.0 | 1.1 | 1.5 | 4.0 | 8.2 | 65.2 | 1.0 | 1.3 | 2.5 | 9.3 | 100.0 | |
| gb | 1.0 | 1.2 | 1.6 | 2.3 | 9.7 | 1.0 | 1.2 | 1.6 | 2.5 | 17.1 | 1.0 | 1.2 | 1.5 | 2.2 | 8.7 | |

---

## Author Comment (AC2) · 16 Dec 2016

**Response to the comment of Anonymous Referee # 2**

*We would like to thank the reviewer for his detailed assessment of the manuscript. Below is our response to the issues raised in the review. The original comment is printed in plain font, our response is printed in italics.*

The manuscript describes the 2015 streamflow drought event relative to the 2003 event based on observed low flow conditions derived over a set of stations. Results are compared also with the corresponding 2015 meteorological drought event analysed in detail in a companion work. I find that the analysis presented provides limited advances for a better understanding of hydrological drought processes and many parts of the manuscript are too qualitative and descriptive.

*The question of advanced understanding based on descriptive statistics and analysis is certainly an important one to assess the significance and potential impact of a study such as ours. We would like to argue that the analysis presented does indeed add to a better understanding of the characteristics of hydrological drought at the large cross-country scale in Europe. The scope of the work was to study a major hydrological event on the European scale soon after its occurrence. Unlike climatological information, the timely observational-based analysis of a hydrological event at a pan-European scale (across country boundaries) has not previously been undertaken due to the lack of hydrological data. Streamflow data are commonly only available in national databases in near-real time and updated large-scale databases would have a significant lag in the order of years between each update. The study thus presents a unique community effort and opportunity to capitalize on our common knowledge and enhanced local detailed information. We have explicitly chosen an approach based on extreme value statistics and seasonality indices, which allows us to interpret processes from spatio-temporal patterns directly, with a minimum number of modeling steps and assumptions. Interpreting patterns of indices and process indicators is often regarded to be superior to classification techniques and modeling because of the minimum number of assumptions made (Grayson and Blöschl, 2001; Laaha and Bloschl, 2006). In the revision, we will further highlighted the added value of our work using additional quantitative analysis (specified in our detailed responses below).*

My major concerns are:
– The influence of antecedent moisture conditions on drought developments is interesting and novel aspect, maybe the most relevant in the work. However it is analysed only on two stations. I suggest the authors to extent this investigation on the whole set of data to derive their conclusions in a more robust manner and spatially over the domain. The use of cluster analysis (or similar more objective techniques) to group stations with similar hydro-meteorological response may be an option. More details on the characteristics of antecedent moisture conditions, for instance timing and magnitude of antecedent precipitation that may reduce the probability of subsequent extreme drought events, would be relevant as well in view of an enhanced predictability and monitoring of low flow conditions.

*The regional perspective is indeed important, hence it was also analyzed and discussed in our paper. An inductive approach has been chosen which infers the significance of the timing of low flow events from antecedent catchment conditions from single example catchments. Seasonality maps (Figure 6) of the relative timing of events were employed to analyze the regional perspective and to generalize the finding to the Pan-European scale. From pattern similarity between onset and severity of the low flow event, we deduced that the seasonality of the onset, being an indicator of antecedent conditions, is clearly related to drought severity at the regional scale.*

*To underpin the general relevance of our finding, and in accordance with the referee comment, we will conduct an additional functional cluster analysis of standardized hydrographs to generalize the local fingerprints provided by the hydrographs of two catchments at the European scale. The clusters will be interpreted with respect to spatio-temporal patterns of low flow and drought indices. In the supplement Fig. S2 we have added draft maps of the clustering to illustrate the value of the additional analysis.*

*We agree that more details on the characteristics of antecedent moisture conditions, for instance timing and magnitude of antecedent precipitation that may reduce the probability of subsequent extreme drought events, would be relevant and provide a view of an enhanced predictability and monitoring of low flow conditions. In the supplement (Fig. S1) we have added maps of the standardized precipitation index (different aggregation intervals were tested) that summarize antecedent conditions from meteorological measurements (ref. reply to Reviewer#1). The maps will be compared to the seasonality indices used to summarize antecedent conditions based on streamflow observations. Note that soil moisture measurements were not available to undertake similar analysis.*

– Section 5.2 is too descriptive and qualitative, and it does not add relevant new knowledge. Furthermore, it suffers of a poor methodological approach. There have been developed automated research algorithms to collect events and information from web and media in a systematic manner. I suggest the authors to implement such methods or to remove completely this section.

*We wish there were automated methods, but they do not exist. The US Drought impact reporter and the European drought impact report inventory (EDII) differ in the way they collect data, but each entry is moderated, i.e. manually checked and coded into the system and manually transcribed as it is not legally possible otherwise to store the data. The JRC media monitor is only a real-time tool, which provides many false hits and so far has not been used in any quantitative analyses due to these difficulties. In reality, this process is not at all automated and data for 2015 does not yet exist as a consolidated dataset. For the revision of the paper we see two options, (a) delete the section, (b) improve the section by citing some key impact reports as anecdotal evidence and discuss the ways forward and difficulties of a more comprehensive impact report collection from web and media better. We clearly prefer option b) but will await the Editor's decision on the issue. We would further like to remark that the EDII has received considerable international attention for its relevance and being a first effort to collect impact data consistently for different sectors at the pan-European scale, see e.g.*

*http://www.nat-hazards-earth-syst-sci.net/16/801/2016/nhess-16-801-2016-discussion.html*

*http://www.hydrol-earth-syst-sci.net/20/2779/2016/hess-20-2779-2016-discussion.html*

*As one reviewer for the last paper above writes: "The fact that systematic drought impact collection is sorely lacking, or non-existent in many cases, illustrates the need for more resources to be directed at such efforts moving forward as a way of establishing a baseline for how we have been, are and will be affected by future droughts in a changing climate. The lack of a long, comprehensive record of impacts is not the fault of the authors and in fact the development and maintenance of the EDII moving forward is critical for future works like this."*

– Description of methods needs to be improved. In particular it is not specified what time series is used to derive fitting functions and return periods. I suppose the reference period, but this should be better clarified.

*We will clarify the issues raised and carefully review and improve the method section.*

In the comparison with meteorological droughts the following references may be relevant.
Bachmair et al., 2016 (http://onlinelibrary.wiley.com/doi/10.1002/wat2.1154/full)
Van Loon and Laaha, 2015
(http://www.sciencedirect.com/science/article/pii/S0022169414008543).
Barker et al., 2016 (http://www.hydrol-earth-syst-sci.net/20/2483/2016/),

*These recently published studies will be considered in the revision*

Minor comments:
Page 1, line 29: please, consider to remove "in this second paper"
*Sentence will be rephrased.*

Page 1, line 30: stream gauge stations instead of records?
*We prefer to keep the term streamflow records as the second part of the sentence refers to records and not to stations.*

Page 2, line 11: please, add the relevant references to support this sentence
*Reference will be added.*

Page 2, line 16: "Droughts… to analyse", too vague, consider to rephrase or remove.
*Sentence will be rephrased.*

Page 2, line 19-20: This concept needs to be better expressed.
*Sentence will be rephrased.*

Page 3, line 1: move the reference to the end of the sentence.
*Done*

Page 3, line 8: please do not abbreviate South, North, East and West throughout the manuscript. Such abbreviation is not a standard.
*They are indeed contained in the list of common abbreviations in Oxford English Dictionary.*

Page 5 lines 9-13: this information is not relevant for this work, consider removing it.
*We suggest to shorten, rather than delete, the description of the software packages.*

Page 5, line 15: the 2013 is also compared to the 2003 event, this should be clarified.
*We will add "(…and relative to the year 2003)"*

Page 5 lines 20-25: I would suggest to synthesize this content and move to the next sections for a better organization of the text and to avoid redundancy. Are the low-flow indices calculated for the 2015, 2003 and reference period? Please, clarify.
*We will clarify this paragraph and consider its placement in the text.*

Page 6, line 18: how do you define "totally recovered", please clarify.
*We will clarify this paragraph.*

Page 6, Section 3.2. Please clarify on what time series you estimate the fitting functions to derive return periods for reference period, 2015 and 2003. Why did you use such fitting functions instead of generalized extreme value or pareto distribution?
*We have described the standard approach of low flow and drought frequency analysis. As this is standard in low flow hydrology, we don't believe it necessary to elaborate. However, we have clarified the data used by extending the text of step (1): "Sample the annual extremes series AES" with "from daily discharge records of the reference period".*

Page 7 line 21: contrasting response instead of dipole?
*We will change the text accordingly.*

Page 7, line 23: the patter discussed seems not including North-Austria. Maybe, because the graphical representation is not very clear in colours and symbols. I strongly suggest improving figures with maps by showing more contrasted colours.
*The figure was carefully designed for the Pan-European scale, and hence can be difficult to read at a local scale. We will do our best to increase the readability of the figures using more contrasting colours.*

Page 7, line 26: the 2015 drought-affected area? Clarify
*We will add a definition of what is meant by 'drought affected area' in the paper referred to.*

Page 8, line 10: Please, clarify.
*We will modify the sentence: "...with the largest deficits occurring in S-Germany, west of the area with lowest flows..."*

Page 9, line 3-4: Consider to remove winter plots from the graph, they do not add relevant information.
*Although this group of stations is not of prime relevance for the paper we prefer to keep the winter boxplots, for the sake of completeness of the analysis.*

Page 9, line 13: low flow threshold? I understood that you were looking at the minimum flows here. Please clarify.
*Thanks, this is a typo and will be corrected to "average annual low flow discharge".*

Page 10 Section 4.5 please, rephrase without using bullet points.
*Done.*

Page 14 line 7: the extreme is most extreme... please rephrase.
*We will change to "when the extreme event was most extreme"*

Page 14 line 25: add "streamflow" to drought.
*Done (this is p14 line 31 in our original document).*

Page 15, line 10-11. Is there any additional drought self-propagation mechanism linked to land atmosphere interactions that could contribute in explaining these processes? Dry soils may lead to lower probability of precipitation and thus cause intensified droughts. See for instance Senevitarne et al. 2010 (Earth-Science Reviews 99 (2010) 125–161).
*This aspect will be considered and a remark added to the paper.*

Page 15 lines 20-33. This text is very speculative and not related to the work presented, please

consider removing it.

*We will carefully evaluate the paragraph and its relevance for the paper and accordingly revise the text.*

**References:**

*Grayson, R. and Blöschl, G., Eds.: Spatial patterns in catchment hydrology: observations and modelling, Cambridge University Press, Cambridge, U.K. ; New York., 2001.*

*Laaha, G. and Bloschl, G.: A comparison of low flow regionalisation methods–catchment grouping, J. Hydrol., 323(1–4), 193–214, 2006.*

Blauhut, V., Stahl, K., Stagge, J. H., Tallaksen, L. M., De Stefano, L., and Vogt, J. (2016): Estimating drought risk across Europe from reported drought impacts, drought indices, and vulnerability factors, Hydrol. Earth Syst. Sci., 20, 2779-2800, doi:10.5194/hess-20-2779-2016, 2016.

Stahl, K., Kohn, I., Blauhut, V., Urquijo, J., De Stefano, L., Acácio, V., Dias, S., Stagge, J. H., Tallaksen, L. M., Kampragou, E., Van Loon, A. F., Barker, L. J., Melsen, L. A., Bifulco, C., Musolino, D., de Carli, A., Massarutto, A., Assimacopoulos, D., and Van Lanen, H. A. J. (2016) Impacts of European drought events: insights from an international database of text-based reports. *Nat. Hazards Earth Syst. Sci.*, 16, 801-819, doi:10.5194/nhess-16-801-2016.

**Figures**

[Figure]

*Figure S1. SPI6 for January 2003 (left) and January 2015 (right). Reference period 1971-2000.*

**2003**

[Figure]

*Figure S2. Functional clustering of the low flow event of 2003 based on monthly standardized streamflow index values SSI of the Jan – Oct period. a) combined cluster map showing allocation of catchments to the clusters, b) combined map of functional models of each cluster, c – h) cluster component maps, i– n) synoptic plots of standardized monthly hydrographs of Cluster 1 – 6 (thin black lines) together with the functional model of each cluster center (bold colored line). Altschlaining is marked by a bold black line, Imbach is marked by a dashed black line.*

**2015**

[Figure]

*Figure S3. Functional clustering of the low flow event of 2015 based on monthly standardized streamflow index values SSI of the Jan – Oct period. a) combined cluster map showing allocation of catchments to the clusters, b) combined map of functional models of each cluster, c – h) cluster component maps, i– n) synoptic plots of standardized monthly hydrographs of Cluster 1 – 6 (thin black lines) together with the functional model of each cluster center (bold colored line). Altschlaining is marked by a bold black line, Imbach is marked by a dashed black line.*

---

## Author Response (AR1)

Dear Prof. Laurent Pfister,

We thank the editor and two referees for their assessment of our manuscript. Please find our detailed responses below. We performed textual edits to the entire paper to increase readability and sharpen the text. We believe we have addressed all points raised by the reviewers carefully and modified the manuscript accordingly.

Kind regards,
Gregor Laaha

**Response to the comment of Anonymous Referee # 1**

*We would like to thank the reviewer for his kind assessment of the manuscript. Below is our response to the issues raised in the review. The original comment is printed in plain font, our response is printed in italics.*

General:
The paper is well written; it is highly descriptive and provides relevant facts on impacts of droughts in different countries for the year 2015. However for the reader it would have been easier to if the different aspects of droughts would have been summarized in a table for the different countries. One aspect however is missing when talking about droughts for a year. It is very important to provide information about the meteorological conditions of the previous year, especially if you look at the status of the aquifers.

*We thank the referee for this excellent evaluation.*
*We added an Appendix where we provide tables summarizing the different aspects of drought by country, for all stations (Table A1) and stratified into stations with predominant summer low flow regime (Table A2) and stations with a predominant winter low flow regime. We further agree that it is relevant to provide information about the meteorological conditions preceding the event. Preconditions were summarized by SPI indices of various aggregation scales (6, 9 and 12 month), and these characteristics were integrated in the analyses. We further assessed to what degree these indices are correlated with low flow magnitude, and how these correlations compare to the correlations of the relative onset index.*

Conclusions:
A European network experts on water scarcity and droughts produced a report in 2007 on drought management plans as part of the Common Implementation Strategy of the Water Framework Directive, which was endorsed by the member States in November 2007. The report sets out recommendations in preparing drought management plans. Examples are shown for some Member States. The authors could have explored in the first place the status of the implementation in the Member States instead of referring to the example given by GWP and WMO.

*We appreciate the reminder and refer to the report and the need for a range of hydrological flow indices in introduction and discussion. The status of RBMP implementation appears not directly relevant to this study's objective of understanding the physical hydrological aspects of extent and propagation of the hydrological drought, but we agree that the report helps to highlight the need and potential benefits of common indices.*

*Reference:*
*Water Scarcity and Droughts Expert Network: Drought Management Plan Report - Including Agricultural, Drought Indicators and Climate Change Aspects. [online] Available from: http://ec.europa.eu/environment/water/quantity/pdf/dmp_report.pdf (Accessed 30 November 2016), 2007.*

**Response to the comment of Anonymous Referee # 2**

*We would like to thank the reviewer for his detailed assessment of the manuscript. Below is our response to the issues raised in the review. The original comment is printed in plain font, our response is printed in italics.*

The manuscript describes the 2015 streamflow drought event relative to the 2003 event based on observed low flow conditions derived over a set of stations. Results are compared also with the corresponding 2015 meteorological drought event analysed in detail in a companion work. I find that the analysis presented provides limited advances for a better understanding of hydrological drought processes and many parts of the manuscript are too qualitative and descriptive.

*The question of advanced understanding based on descriptive statistics and analysis is certainly an important one to assess the significance and potential impact of a study such as ours. We would like to argue that the analysis presented does indeed add to a better understanding of the characteristics of hydrological drought at the large cross-country scale in Europe. The scope of the work was to study a major hydrological event on the European scale soon after its occurrence. Unlike climatological information, the timely observational-based analysis of a hydrological event at a pan-European scale (across country boundaries) has not previously been undertaken due to the lack of hydrological data. Streamflow data are commonly only available in national databases in near-real time and updated large-scale databases would have a significant lag in the order of years between each update. The study thus presents a unique community effort and opportunity to capitalize on our common knowledge and enhanced local detailed information. We have explicitly chosen an approach based on extreme value statistics and seasonality indices, which allows us to interpret processes from spatio-temporal patterns directly, with a minimum number of modeling steps and assumptions. Interpreting patterns of indices and process indicators is often regarded to be superior to classification techniques and modeling because of the minimum number of assumptions made (Grayson and Blöschl, 2001; Laaha and Bloschl, 2006). In the revision, we further highlighted the added value of our work using additional quantitative analysis (specified in our detailed responses below).*

My major concerns are:
– The influence of antecedent moisture conditions on drought developments is interesting and novel aspect, maybe the most relevant in the work. However it is analysed only on two stations. I suggest the authors to extent this investigation on the whole set of data to derive their conclusions in a more robust manner and spatially over the domain. The use of cluster analysis (or similar more objective techniques) to group stations with similar hydro-meteorological response may be an option. More details on the characteristics of antecedent moisture conditions, for instance timing and magnitude of antecedent precipitation that may reduce the probability of subsequent extreme drought events, would be relevant as well in view of an enhanced predictability and monitoring of low flow conditions.

*The regional perspective is indeed important, hence it was also analyzed and discussed in our paper. An inductive approach has been chosen which infers the significance of the timing of low flow events from antecedent catchment conditions from single example catchments. Seasonality maps (Figure 6) of the relative timing of events were employed to analyze the regional perspective and to generalize the finding to the Pan-European scale. From pattern similarity between onset and severity of the low flow event, we deduced that the seasonality of the onset, being an indicator of antecedent conditions, is clearly related to drought severity at the regional scale.*

*To underpin the general relevance of our finding, and in accordance with the referee comment, we conducted an additional functional cluster analysis to generalize the local fingerprints provided by the hydrographs of two catchments at the European scale. The results are presented in new Section 4.5 which includes two new figures (Fig. 6 and 7).*

*We agree that more details on the characteristics of antecedent moisture conditions, for instance timing and magnitude of antecedent precipitation that may reduce the probability of subsequent extreme drought events, would be relevant and provide a view of an enhanced predictability and monitoring of low flow conditions. We therefore extended the analysis by including standardized precipitation indices (different aggregation intervals were tested) that summarize antecedent conditions based on meteorological measurements (ref. reply to Reviewer#1). The analyses include maps of January-SPI6, which we consider notably relevant for characterizing preconditions of the studied events (new Fig. 8). Hereby, the scope of former Section 4.5 on the "Effect of seasonality" was extended, and the title was changed to "Effect of preconditions" (now section 4.6). We also conducted a correlation analysis to assess the relative information content of the various indices of antecedent conditions (SPIs and relative onset of hydrological drought) for predicting summer low flow conditions. Note that soil moisture measurements were not available to undertake similar analysis.*

− Section 5.2 is too descriptive and qualitative, and it does not add relevant new knowledge. Furthermore, it suffers of a poor methodological approach. There have been developed automated research algorithms to collect events and information from web and media in a systematic manner. I suggest the authors to implement such methods or to remove completely this section.

*We wish there were automated methods, but they do not exist. The US Drought impact reporter and the European drought impact report inventory (EDII) differ in the way they collect data, but each entry is moderated, i.e. manually checked and coded into the system and manually transcribed as it is not legally possible otherwise to store the data. The JRC media monitor is only a real-time tool, which provides many false hits and so far has not been used in any quantitative analyses due to these difficulties. In reality, this process is not at all automated and data for 2015 does not yet exist as a consolidated dataset.*
*We agree that the section content deviates much from the main approach of our study, hence we removed Section 5.2, considerably shortened the discussion of impacts and integrated some of the text with Section 5.4 (now reordered to 5.3).*

− Description of methods needs to be improved. In particular it is not specified what time series is used to derive fitting functions and return periods. I suppose the reference period, but this should be better clarified.

*We clarified the issues raised and carefully reviewed and improved the method section.*

In the comparison with meteorological droughts the following references may be relevant.
Bachmair et al., 2016 (http://onlinelibrary.wiley.com/doi/10.1002/wat2.1154/full)
Van Loon and Laaha, 2015
(http://www.sciencedirect.com/science/article/pii/S0022169414008543).
Barker et al., 2016 (http://www.hydrol-earth-syst-sci.net/20/2483/2016/),

*These recently published studies were considered in the revision*

Minor comments:

Page 1, line 29: please, consider to remove "in this second paper"
*Sentence was rephrased.*

Page 1, line 30: stream gauge stations instead of records?
*We prefer to keep the term streamflow records as the second part of the sentence refers to records and not to stations.*

Page 2, line 11: please, add the relevant references to support this sentence
*Reference was added.*

Page 2, line 16: "Droughts… to analyse", too vague, consider to rephrase or remove.
*Sentence was removed.*

Page 2, line 19-20: This concept needs to be better expressed.
*Sentence was rephrased to better express the concept.*

Page 3, line 1: move the reference to the end of the sentence.
*The sentence was rephrased and the reference moved.*

Page 3, line 8: please do not abbreviate South, North, East and West throughout the manuscript. Such abbreviation is not a standard.
*They are indeed contained in the list of common abbreviations in Oxford English Dictionary. We find the abbreviations useful to increase readability.*

Page 5 lines 9-13: this information is not relevant for this work, consider removing it.
*We shortened the description of the software packages.*

Page 5, line 15: the 2013 is also compared to the 2003 event, this should be clarified.
*Done.*

Page 5 lines 20-25: I would suggest to synthesize this content and move to the next sections for a better organization of the text and to avoid redundancy. Are the low-flow indices calculated for the 2015, 2003 and reference period? Please, clarify.
*We have clarified in Line 20: "A comprehensive characterisation of hydrological drought events, such as those of 2015 and 2003, requires a number of different indices …*

Page 6, line 18: how do you define "totally recovered", please clarify.
*We changed the sentence accordingly: "…, and two droughts are pooled if the catchment store has not totally recovered from the first drought when the second drought episode begins."*

Page 6, Section 3.2. Please clarify on what time series you estimate the fitting functions to derive return periods for reference period, 2015 and 2003. Why did you use such fitting functions instead of generalized extreme value or pareto distribution?
*We have described the standard approach of low flow and drought frequency analysis. As this is standard in low flow hydrology, we don't believe it necessary to elaborate. However, we have clarified the data used by extending the text of step (1): "Sample the annual extremes series AES" with "from daily discharge records of the reference period". We also made minor changes to the remaining steps of the approach.*

Page 7 line 21: contrasting response instead of dipole?

*We changed the text accordingly.*

Page 7, line 23: the patter discussed seems not including North-Austria. Maybe, because the graphical representation is not very clear in colours and symbols. I strongly suggest improving figures with maps by showing more contrasted colours.
*The figure was carefully designed for the Pan-European scale, and hence can be difficult to read at a local scale. We did our best to increase the contrasts in the maps.*

Page 7, line 26: the 2015 drought-affected area? Clarify
*We clarified: "the area affected by the hydrological drought".*

Page 8, line 10: Please, clarify.
*We changed the wording: "...with the largest deficits occurring in S-Germany, west of the area with lowest flows..."*

Page 9, line 3-4: Consider to remove winter plots from the graph, they do not add relevant information.
*Although this group of stations is not of prime relevance for the paper we prefer to keep the winter boxplots, for the sake of completeness of the analysis.*

Page 9, line 13: low flow threshold? I understood that you were looking at the minimum flows here. Please clarify.
*Thanks, this is a typo and was corrected to "average annual low flow discharge".*

Page 10 Section 4.5 please, rephrase without using bullet points.
*Done.*

Page 14 line 7: the extreme is most extreme… please rephrase.
*The paragraph was entirely rephrased.*

Page 14 line 25: add "streamflow" to drought.
*Done (this is p14 line 31 in our original document).*

Page 15, line 10-11. Is there any additional drought self-propagation mechanism linked to land atmosphere interactions that could contribute in explaining these processes? Dry soils may lead to lower probability of precipitation and thus cause intensified droughts. See for instance Senevitarne et al. 2010 (Earth-Science Reviews 99 (2010) 125–161).
*This aspect was considered and a remark added to the paper.*

Page 15 lines 20-33. This text is very speculative and not related to the work presented, please consider removing it.
*Despite that the entire section was deleted, we carefully evaluated the paragraph and its relevance for the paper. Whilst perhaps not directly related to the specific analysis, the text is not speculative, and parts of the text were kept and integrated in the subsequent section.*

**References:**

*Grayson, R. and Blöschl, G., Eds.: Spatial patterns in catchment hydrology: observations and modelling, Cambridge University Press, Cambridge, U.K. ; New York., 2001.*

[revised manuscript text omitted]
 | 1.0 | 1.2 | 1.4 | 1.8 | 4.9 | 1.0 | 1.1 | 1.2 | 2.9 | 4.6 | 1.1 | 1.1 | 1.3 | 2.6 | 23.4 | |

---

## Referee Report (RR1)

I have appreciated the efforts made by authors to clarify my major comments.
I strongly suggest that a native English speaker revises the writing, the current version of the manuscript has several errors and does not flows well.

I have still some minor comments
Line 27. Large part of Europe was …
Line 27. Footprint. Maybe in the abstract would be better to use a more widely used terminology.
Line 28. I do not see the reason to mention in brackets "magnitude", here and in the following lines of abstract. Consider to remove it.
Line 37. Diverging. I think it is not clear to what it is referring to.

Line 13. Remove now, not needed
Line 30. I think "timely" is more appropriate than "thorough" in this context

Line 24. "and are therefore spatially variable as well". Consider to remove it, not needed.

Line 28. with respect to their spatial coverage

Line 2-4. You do not need to introduce the general formulation with n
Lines 7-10. Information not relevant, consider to remove it

Line 8. Suggested instead of recommended.
Line 14. I suppose return period have been calculated by inversion of the fitting function. Plase clarify.
Line 28. Mixed mixture model… are you sure? Strange nomenclature
Line 29, Please, clarify for what event has been computed.

Line 6. Please, introduce in methods how you identify winter low flows.
Line 17. Avoid the use of slightly, repetition
Line 22. Exceptional conditions. Not clear what you mean, please clarify.
Line 30 and following. This material should be mentioned in the method section.

Line 2. Unclear, please clarify.

Lines 23-26. Material for methods

Line 12. While some regional features…
Line 20. The maps exhibit … features. Please, consider to remove, not needed.
Line 24-25. Rephrase

Lines 1-10. Please, clarify why you have performed this analysis and what are the main findings. Not clear at all. Furthermore, this mostly material for methods.
Line 4. Clarify to what correlations these numbers refer.
Lines 5-6. Counterintuitive result, please clarify.
Line 14. Please, add reference "droughts is one of the most costly hazards"

Line 18. It is interesting to analyse? Let the reader think if it is interesting or not.

Page 16. I still think that the paper would benefit if this part would be removed completely. This is not related to your findings, It does not add any relevant information.
Line 33. Personal communications by who?? Please, use appropriate references.

Page 23. First two references are reported wrongly.

Figure 2. It would be useful to see the number of stations used for the boxplots
Figure 3. Add color palette
Figure 6 and 7. Panel a and b are redundant, please remove them. In panels c-h, use larger dots. For the reference stations of altschtaining use a different type of line, is confounded with the y=0 line. I suppose that each gray line is a station in panels i-n. Please clarify in captions.
Figure 10. Exaplain the meaning of seasonality in caption. Change colour for potentially ongoing (not extreme) and potentially ongoing (most extreme) to improve visual distinction.
Table A1-A3. I suppose that numbers refer to return period. Please clarify in caption. Bars on the right are redundant and do not add relevant information, please consider to remove them.

---

## Author Response (AR2)

*We thank the reviewer for his very detailed comments; we have addressed them all in the final revision of our MS. Please find our responses in italic below the referee comments.*

I have appreciated the efforts made by authors to clarify my major comments.
I strongly suggest that a native English speaker revises the writing, the current version of the manuscript has several errors and does not flows well.
*Done*

I have still some minor comments
Line 27. Large part of Europe was …
*We prefer to keep it unchanged*
Line 27. Footprint. Maybe in the abstract would be better to use a more widely used terminology.
*Definition was added*
Line 28. I do not see the reason to mention in brackets "magnitude", here and in the following lines of abstract. Consider to remove it.
*There are different severity measures of drought events around, hence this specification is useful.*
Line 37. Diverging. I think it is not clear to what it is referring to.
*Done*

Line 13. Remove now, not needed
*Done*
Line 30. I think "timely" is more appropriate than "thorough" in this context
*Done*

Line 24. "and are therefore spatially variable as well". Consider to remove it, not needed.
*We prefer to keep it.*

Line 28. with respect to their spatial coverage
*Done*

Line 2-4. You do not need to introduce the general formulation with n
*Done*
Lines 7-10. Information not relevant, consider to remove it
*There are actually diverging uses of drought definitions, and we need to be crystal-clear about the method we use.*

Line 8. Suggested instead of recommended.
*We see it more as a recommendation.*
Line 14. I suppose return period have been calculated by inversion of the fitting function. Plase clarify.
*Done*
Line 28. Mixed mixture model… are you sure? Strange nomenclature
*Yes, it's the proper name of the method.*

Line 29, Please, clarify for what event has been computed.
*We clarified in line 20 and line 29.*

Line 6. Please, introduce in methods how you identify winter low flows.
*We added the definition on page 6, line 28: We further distinguish between summer (May –
November) and winter (December – April) low flow season, and classify gauges according to
their dominant low flow season into summer and winter regimes.*
Line 17. Avoid the use of slightly, repetition
*Done*
Line 22. Exceptional conditions. Not clear what you mean, please clarify.
*Done*
Line 30 and following. This material should be mentioned in the method section.
*Done, it was moved to Section 3.2*

Line 2. Unclear, please clarify.
*The passage has been removed.*

Lines 23-26. Material for methods
*The sentence was moved to Section 4.5.*

Line 12. While some regional features…
*Done*
Line 20. The maps exhibit … features. Please, consider to remove, not needed.
*Done*
Line 24-25. Rephrase
*Done*

Lines 1-10. Please, clarify why you have performed this analysis and what are the main
findings.
Not clear at all. Furthermore, this mostly material for methods.
Line 4. Clarify to what correlations these numbers refer.
Lines 5-6. Counterintuitive result, please clarify.
Line 14. Please, add reference "droughts is one of the most costly hazards"
*Done*

Line 18. It is interesting to analyse? Let the reader think if it is interesting or not.
*We modified the sentence.*

Page 16. I still think that the paper would benefit if this part would be removed completely.
This is not related to your findings, It does not add any relevant information.
*We carefully evaluated the paragraph and its relevance for the paper. We believe the
provided information is very relevant for drought management and for the discussion whether
universal or specific indices are needed. Hence, we prefer to keep it in the paper.*Line 33.
Personal communications by who?? Please, use appropriate references.

Page 23. First two references are reported wrongly.
*Done*

Figure 2. It would be useful to see the number of stations used for the boxplots
Figure 3. Add color palette
*It is indicated in the legend that colour codes are those of Fig. 1. For the sake of a optimal layout we prefer to leave it unchanged.*

Figure 6 and 7. Panel a and b are redundant, please remove them. In panels c-h, use larger dots.
*It is right that the lines and points are also shown in individual panels, but the synoptic view provides additional, complementary information to the individual plots. We therefore prefer to keep panels a and b. In panels c-h and a we increased the symbol size.*
For the reference stations of altschtaining use a different type of line, is confounded with the y=0 line. I suppose that each gray line is a station in panels i-n. Please clarify in captions.
*Line signature of Altschlaining was changed and grey lines were explained in the caption.*

Figure 10. Exaplain the meaning of seasonality in caption. Change colour for potentially ongoing (not extreme) and potentially ongoing (most extreme) to improve visual distinction.
*Done*

Table A1-A3. I suppose that numbers refer to return period. Please clarify in caption. Bars on the right are redundant and do not add relevant information, please consider to remove them.
*We added this information.*